# Role of atmospheric rivers in shaping long term Arctic moisture variability

Zhibiao Wang [1], Qinghua Ding [2] ✉, Renguang Wu [3], Thomas J. Ballinger [4] ✉, Bin Guan [5], Deniz Bozkurt [6,7,8], Deanna Nash[9], Ian Baxter [2], Dániel Topál[10,11], Zhe Li [2], Gang Huang [12], Wen Chen [13], Shangfeng Chen [1], Xi Cao [1] & Zhang Chen[14]

Atmospheric rivers (ARs) reaching high-latitudes in summer contribute to the majority of climatological poleward water vapor transport into the Arctic. This transport has exhibited long term changes over the past decades, which cannot be entirely explained by anthropogenic forcing according to ensemble model responses. Here, through observational analyses and model experiments in which winds are adjusted to match observations, we demonstrate that low-frequency, large-scale circulation changes in the Arctic play a decisive role in regulating AR activity and thus inducing the recent upsurge of this activity in the region. It is estimated that the trend in summertime AR activity may contribute to 36% of the increasing trend of atmospheric summer moisture over the entire Arctic since 1979 and account for over half of the humidity trends in certain areas experiencing significant recent warming, such as western Greenland, northern Europe, and eastern Siberia. This indicates that AR activity, mostly driven by strong synoptic weather systems often regarded as stochastic, may serve as a vital mechanism in regulating long term moisture variability in the Arctic.

Arctic surface air temperatures have shown a warming trend at a rate more than twice that of the global average in recent decades, attributed to various Arctic Amplification (AA) processes driven by both anthropogenic and natural climate forcing[1–9]. As constrained by the Clausius–Clapeyron (CC) relationship, Arctic atmospheric warming also leads to atmospheric moistening, resulting in higher specific humidity, greater cloud cover and cloud water content, and more

precipitation across the Arctic[10–13]. This moisture increase has substantially altered the Arctic hydrological and cryospheric variability over the past few decades[14,15], owing to various moisture-related positive feedbacks connected to changing radiative properties of the atmosphere, clouds, and surface conditions.

This increase in moisture is prominent throughout the year, with the most significant rise occurring during the summer months

[1]Center for Monsoon System Research, Institute of Atmospheric Physics, Chinese Academy of Sciences, Beijing, China. [2]Department of Geography and Earth Research Institute, University of California, Santa Barbara, Santa Barbara, CA, USA. [3]School of Earth Sciences, Zhejiang University, Hangzhou, China. [4]International Arctic Research Center, University of Alaska Fairbanks, Fairbanks, AK, USA. [5]Joint Institute for Regional Earth System Science and Engineering, University of California, Los Angeles, Los Angeles, CA, USA. [6]Department of Meteorology, University of Valparaíso, Valparaíso, Chile. [7]Center for Climate and Resilience Research (CR)2, Santiago, Chile. [8]Center for Oceanographic Research COPAS COASTAL, University of Concepción, Concepción, Chile. [9]Center for Western Weather and Water Extremes, Scripps Institution of Oceanograph, University of California San Diego, La Jolla, CA, USA. [10]Earth and Climate Research, Earth and Life Institute, Université catholique de Louvain, Louvain-la-Neuve, Belgium. [11]Institute for Geological and Geochemical Research, HUN-REN Research Centre for Astronomy and Earth Sciences, MTA-Centre for Excellence, Budapest, Hungary. [12]State Key Laboratory of Numerical Modelling for Atmospheric Sciences and Geophysical Fluid Dynamics, Institute of Atmospheric Physics, Chinese Academy of Sciences, Beijing, China. [13]Department of Atmospheric Sciences, Yunnan University, Kunming, China. [14]School of Atmospheric Sciences, Chengdu University of Information Technology, Chengdu, China. ✉e-mail: qinghua@ucsb.edu; tjballinger@alaska.edu

(June–July–August, abbreviated as JJA)[16,17], which has been attributed to multiple sources triggered by global warming, including increased evaporation from local ocean and surrounding continents[18], enhanced sublimation of ice and snow within the Arctic[19,20], and intensified moisture transport from lower latitudes[15,16,21–27]. Climate models forced by historical anthropogenic emissions can replicate a warmer and more humid Arctic in summer, featuring a rather uniform rise in atmospheric temperature and specific humidity across most of the Arctic. Future climate projections suggest that this wetting trend will intensify under continued global warming scenarios, and bring about a number of cascading effects in the Arctic, such as a precipitation regime shift from snow to rain that will substantially change local ecosystems in the coming decades[28–31]. While these modeling studies highlight the dominance of the CC relationship in shaping Arctic temperature and specific humidity, it is evident that over recent decades summertime changes in these parameters are influenced by large-scale circulation variability, manifested as a long-term trend toward the local barotropic high-pressure anomaly situated over Greenland over the past four decades. This summertime circulation trend pattern is thought to have origins, in part, from internal climate variability and contribute to warming in the mid-to-lower troposphere of the Arctic, as well as the melting of sea ice and the Greenland ice sheet (GrIS) through adiabatic warming processes[32–37]. Thus, both anthropogenic forcing and internal variability should be factored in when considering the underlying mechanisms behind recent moisture trends in the Arctic.

While the CC equation governs the temperature–humidity relationship on a global scale, internal atmospheric circulation variability also plays a key role in the Arctic. Circulation variability regulates the distribution and transport of moisture across a wide range of time-scales through its effects on weather systems including extratropical storms such as Arctic cyclones and atmospheric rivers (ARs). ARs are characterized by elongated, ribbon-like plumes of intense water vapor, driven by synoptic-scale cold-frontal systems, which normally move northeastward and persist for a few days in the extratropcis[12,38,39]. ARs typically form as a result of the interaction between a cold front, which transports a substantial amount of water vapor, and a warm front[40]. Thus, ARs represent a unique and intensely extreme phenomenon linked to the storm tracks[41,42], closely tied to extratropical cyclone and anticyclone activity[43].

Over the annual cycle, ARs contribute to more than 90% of poleward transport of water vapor into the Arctic and are observed across the Arctic for about 1–3 days per month throughout the year, with their highest occurrence from June to August (Supplementary Fig. 1a, which can be found in the Supplementary Information section). This is mainly due to the northward shift of the subtropical jet in these warmer months relative to the cold season, resulting in more ARs propagating into the Arctic[44–46] (Supplementary Fig. 1b, c). Thus, boreal summer stands out as a critical period during which ARs have a greater potential of altering moisture distribution in the Arctic and consequently impacting the local climate.

It is worth noting that AR frequency in the Arctic has increased in recent decades, particularly over western Greenland, where stronger ARs have contributed to significant melt over the GrIS' ablation zone[47]. These AR-related changes have complex origins, and their causes remain uncertain. Suggestions have been made that mid- and high-latitude AR characteristics (i.e., frequency, intensity, trajectory and duration) are sensitive to thermodynamic effects related to global warming[42,48–54] and/or anthropogenic and natural aerosol forcing[55–57]. Sufficient moisture and eddy kinetic energy are necessary conditions for the formation and activity of ARs. Under global warming, the available potential energy and atmospheric moisture are expected to increase, particularly in the Arctic, consequently increasing the occurrence of ARs there[58–60]. These AR changes could also be attributed to atmospheric internal variability[61–63]. It is unclear whether a

combination of these mechanisms can fully account for the recent observed features of ARs and the moistening trend in the Arctic. A particularly unaddressed question is how global warming and the low-frequency circulation variability, both separately and in concert, have moistened the Arctic in the past decades through AR activity. This is an emerging topic of interest given the significant potential impacts of summertime ARs in destabilizing the GrIS in recent decades. For instance, extreme GrIS melt in the summer of 2012, which contributed 1.2 mm global mean sea level rise[64], was exacerbated by AR activity[65,66] connected with amplified JJA anticyclonic circulation over the region[67,68]. More recently, ARs prompted a mid-summer melt event in northeast Greenland in 2014 and a late summer melt event across much of the GrIS ablation area in 2021[69].

One hurdle in understanding the interplay between ARs and large-scale circulation change lies in the inability of models in capturing the observed circulation pattern in the Arctic over the past decades when they are forced by anthropogenic radiative forcing or observed boundary SST and sea ice forcing[70–72]. This poses a significant challenge toward precisely investigating the governing role of circulation on ARs. In this study, we overcome this challenge by employing a nudging approach in Community Earth System Model version 1 (CESM1)[73] to examine how ARs respond when observed wind trends are introduced into the model. We then compare these simulated AR changes against observations and AR responses to anthropogenic forcing in the same model. Through this approach, we aim to disentangle the relative roles and underlying mechanisms of anthropogenic forcing and large-scale circulation in regulating the moisture distribution in the Arctic via AR activity. Our goal is to develop physical and qualitative insights about the contribution of ARs to the Arctic moistening trends. We focus our analysis on the summer months (JJA) when the Pan-Arctic moistening trend is most notably positive and significant over the period (Fig. 1b and Supplementary Figs. 1a and 2).

## Results

### Concurrent changes in large-scale atmospheric circulation and AR characteristics

Shaped by the mean atmospheric circulation pattern (Supplementary Fig. 3c), the average frequency of summertime ARs in each grid point is about 3–4 days per month over western Greenland, northern Europe, from Scandinavia to western Russia and northward to the Barents Sea, and eastern Siberia (Supplementary Fig. 3a). Climatologically, specific humidity, circulation, and air temperature in summer exhibit a consistent pattern, with a minimum value centered at the pole surrounded by higher values in the subarctic (Supplementary Fig. 3b–d).

Over the past few decades, coinciding with the period of enhanced AA and rapid global warming, the frequency of JJA AR events reaching the Arctic exhibits an overall increase of 0.1 days month⁻¹ decade⁻¹ from 1979 to 2019. However, this increase is not uniform. AR frequency increases significantly over northern Canada extending to western Greenland, eastern Siberia, and small portions of the central Arctic and northern Europe, while the North Atlantic Ocean, western Alaska, and central Siberia witness weak negative trends (Fig. 1a). Patterns of long-term variations in AR frequency reveal notable similarities to the observed trends in large-scale JJA circulation (Fig. 1c). Upper-level geopotential heights significantly increase across western Greenland, northern Europe, and eastern Siberia, (highlighted in Supplementary Fig. 3e) but over the North Atlantic Ocean, Alaska, and central Siberia the increase is relatively small. In line with these atmospheric circulation trends, both specific humidity and temperature display pronounced rises over western Greenland, northern Europe, and eastern Siberia (Fig. 1b–d).

However, these observed changes are not fully captured by our best estimate of the climate response to anthropogenic forcing, as derived from ensemble means of historical simulations in the Coupled Model Intercomparison Project Phase 6 (CMIP6) and the CESM2 large ensemble. The ensemble mean model simulations show a slight

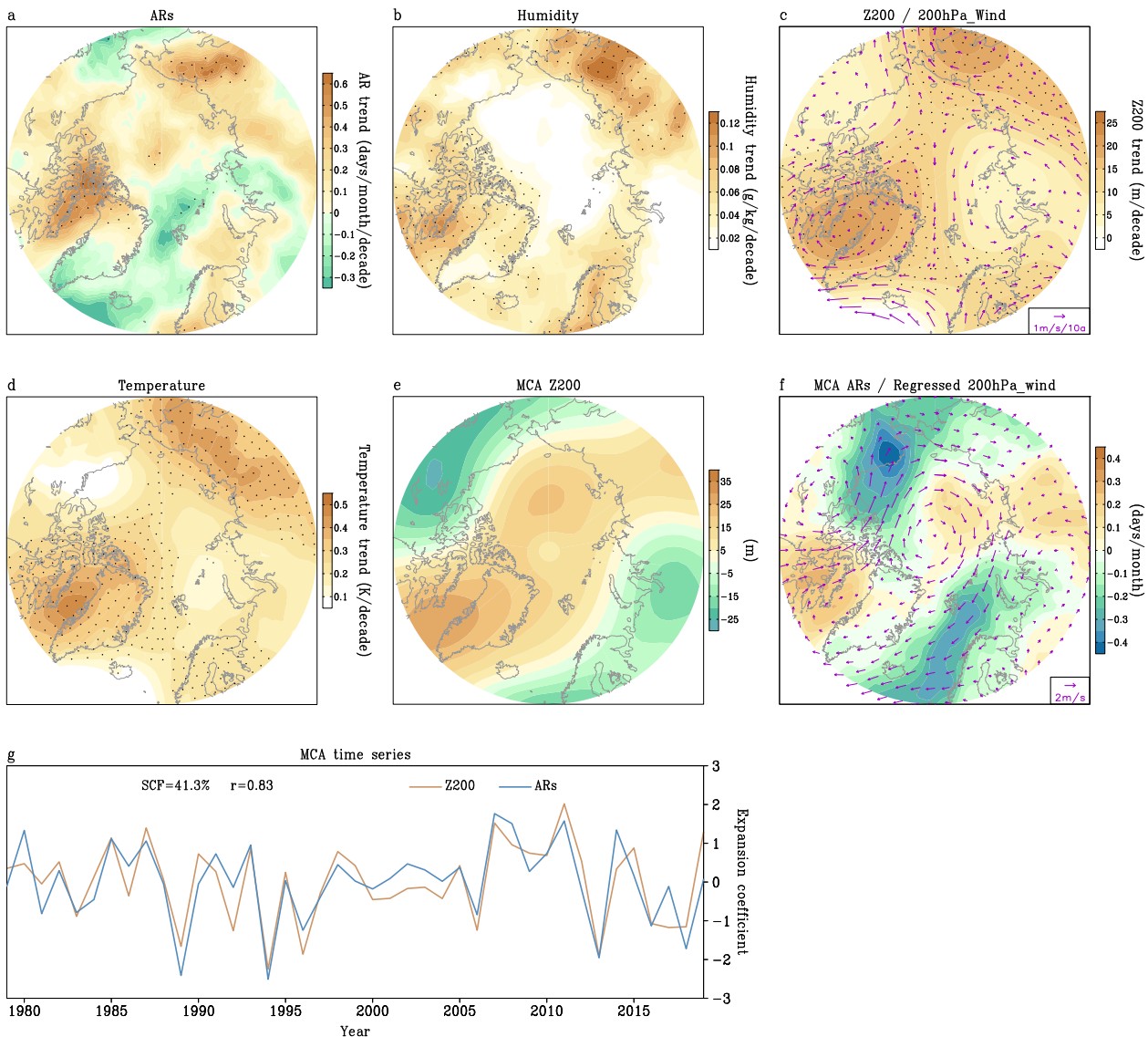

**Fig. 1 | Relationships between summer atmospheric rivers (ARs) and atmospheric variables. a–d** Linear trends of June–July–August (JJA) AR frequency (days/month/decade) (**a**), lower to middle tropospheric (surface to 500 hPa average) specific humidity (g/kg/decade) (**b**), 200 hPa geopotential height (Z200, shaded) (m/decade) and 200 hPa winds (200hPa_Wind, vectors) (m/s/decade) (**c**), and tropospheric (surface to 200 hPa average) air temperature (K/decade) (**d**) in the Arctic from 1979 to 2019. **e, f** Spatial pattern of the leading Maximum Covariance Analysis (MCA) mode of detrended summer Z200 (**e**, in m) and AR frequency (**f**, in days/month) over the period. **g** Standardized time series of the leading MCA mode of summer Z200 (orange line) and AR frequency (blue line). Black dots in (**a–d**) denote statistically significant trends at the 95% confidence level. The scale for the wind trend is shown at the bottom-right corner of (**c**). Wind anomalies (vectors, m/s) at 200 hPa in (**f**) are obtained by linear regression of anomalous JJA winds at 200 hPa against the standardized time series of the Z200 pattern in the leading MCA mode. The scale for wind anomalies is shown at the bottom-right corner of (**f**). "SCF" in (**g**) indicates the squared covariance fraction of the leading MCA mode. "r" in (**g**) indicates the correlation coefficient between the time series of Z200 and AR frequency patterns in the leading MCA mode.

change or nearly uniform rise of geopotential height (200 hPa), air temperature (surface to 200 hPa average), and specific humidity (surface to 500 hPa average) over Greenland and Eurasia, respectively, which strongly contrasts observational findings (Fig. 1 vs. Supplementary Fig. 4) and may signify the role of internal variability in shaping the patterns. It has been extensively discussed in earlier studies that this discrepancy is a common issue across most historical simulations, partially attributed to models' limitations in replicating the observed low-frequency variability of tropical–extratropical teleconnections[33,36,37,71,72,74,75]. In particular, the ensemble mean of JJA ARs displays only an increasing trend, with the highest values located over the Arctic Ocean close to the North Pacific, rather than primarily centered atop and around Greenland as observed (Supplementary Fig. 5a). This suggests that the storm track shift over the North Pacific

close to the Bering Strait has a larger contribution to the increasing trend of ARs in the Arctic under the influence of AA in models.

A better understanding of these discrepancies between observations and the model response to external forcing is necessary to accurately assess the sensitivity of ARs to global warming and to account for other potential forcing factors. The above-described alignment between the observed trends in atmospheric circulation, temperature and specific humidity fields, and AR frequency suggests that large-scale circulation plays a key role in regulating poleward moisture transport through the circulation's control on ARs[76–78] (Fig. 1a, c). We assess these connections from a statistical perspective by employing maximum covariance analysis (MCA)[79] to objectively examine the primary coupled patterns among climatic variables. Figure 1e, f shows the spatial pattern of the leading MCA mode of

detrended JJA 200 hPa geopotential heights (Z200, m) and JJA frequency of ARs between 1979 and 2019. The leading MCA mode (MCA1) explains 41.3% of the squared covariance and its spatial pattern clearly exhibits a zonal wave number 2 structure along 70°N. Regarding the MCA1 pattern, the positive signals of Z200 and ARs are found over western Greenland and the Siberian Arctic, while negative loadings of the two variables are observed over western Europe/western Siberia and areas from western Canada to the Bering Strait. Apparently, these two patterns bear some resemblance to the long-term trends of ARs and circulation, respectively, particularly over Greenland. This resemblance suggests that the secular trends of Z200 and ARs over Greenland may be physically connected, as indicated by their interannual connections identified by MCA1.

MCA1-related 200 hPa wind anomalies demonstrate how JJA mean circulation anomalies shape ARs on interannual timescales. During the summer season, westerly winds prevail in the Arctic troposphere across the region in the climatology (Supplementary Fig. 3c, vectors). ARs are more frequent over the regions marked by strong southerly wind anomalies, while decreased AR frequency is seen over the regions dominated by anomalous JJA northerly or easterly flows (Fig. 1f, indicated by wind vectors). This suggests a strong governing forcing of anomalous seasonal mean flow on AR activities within the season on interannual timescales. In light of the strong impacts of seasonal mean circulation on ARs, if large-scale circulation also varies on low-frequency timescales in specific areas, ARs may respond to these changes and reflect long-term variability.

Moreover, we observe a consistent coupling pattern in the MCA mode between any two fields from Fig. 1, whether trends are removed or not (figures not shown). To further link the change of this MCA mode to other fields of the Arctic climate system, we correlate the time series of MCA1-ARs (or MCA1-Z200) with the detrended JJA low cloud, downward longwave radiation, surface air temperature, and sea ice. We obtain significant correlations among these variables, indicating the importance of the large-scale circulation trend in creating a warmer, cloudier, and more humid atmosphere, characterized by increased AR incursions into many areas over the past decades. This, in turn, amplifies downward longwave radiation and leads to accelerated Arctic sea ice and land ice melt (Supplementary Fig. 6).

## Contribution of ARs to long-term changes in specific humidity

While ARs are statistically connected with large-scale atmospheric variables, such as geopotential height, specific humidity and temperature, in the Arctic on interannual and lower frequency timescales (see Fig. 1), it remains unclear how ARs are physically driven by these large-scale climate variables and potentially feedback on these fields. In this section, we use 6-hourly changes in specific humidity to examine how moisture changes in some regions are associated with 6-hourly AR propagation into the Arctic. This analysis will enable us to develop an approach to estimate the contribution of ARs to the increasing humidity observed over the past 41 years. We first focus our analysis on the western Greenland region, given that it has witnessed the most significant increases in AR frequency and lower tropospheric specific humidity. The year 2012 is particularly noteworthy as it marked the most extreme melt season of the GrIS and Pan-Arctic sea ice since 1979[64]. Therefore, our first examination centers on that year. An MCA analysis between daily AR frequency and specific humidity anomalies during JJA of 2012 is computed to understand their day-to-day connection (for more details, refer to the Methods section). Climatological seasonal cycles and interannual variability are removed from each grid cell to ensure that the identified MCA pattern primarily reflects the connection on shorter timescales.

The leading MCA mode, explaining 58.3% of the squared covariance, displays a large positive AR loading west of Greenland that is collocated with a widespread increase in specific humidity (Fig. 2a, b). The time series of the ARs and moisture modes closely covary

throughout the season (Fig. 2d). In particular, in mid-July, a strong AR preceded a significant surge of atmospheric moisture over most of Greenland[65,66,80] and impacted Summit Station, where exceptional surface air temperatures reached 2.2 °C[81]. This moisture intrusion resulted in widespread ice sheet melt, a phenomenon not observed since 1889[80,82,83]. An MCA analysis for all summers (by connecting all summertime daily data together) from 1979 to 2019 over the region yields a similar coupling pattern (Supplementary Fig. 7a, b). Likewise, MCA for the regions over northern Europe and eastern Siberia exhibits a highly similar coupled mode and a close coherence in the temporal variations for individual summers from 1979 to 2019 (Supplementary Figs. 8 and 9). This highlights the consistent influence of ARs in driving specific humidity changes on synoptic timescales. To illustrate this effect across all summers, we make composites of specific humidity associated with the strongest ARs centered in three key regions (western Greenland, northern Europe, and eastern Siberia, Supplementary Fig. 3e). In each of these regional centers, specific humidity begins to increase approximately 1 day prior to the onset of ARs. Subsequently, following the occurrence of ARs, specific humidity experiences a rapid increase and peaks approximately 1 day after the AR onset, and then swiftly returns to values similar to those observed before the onset of ARs (Fig. 2c and Supplementary Fig. 10). We also create similar composites for areas near the North Pole and southwestern Alaska (Supplementary Fig. 3e), where long-term trends of ARs show a strong increase and decrease over the past four decades, respectively. A similar temporal progression of ARs and specific humidity changes is observed in all of these areas (Supplementary Fig. 11). This analysis suggests that specific humidity changes across most Arctic regions are partly determined by AR activity within the season. Possibly, changes in long-term trends of atmospheric moisture are also mirrored by long-term changes in Arctic AR frequency to some extent.

To quantify the impacts of ARs on moisture changes over the past four decades, we use a statistical method to differentiate AR impacts on the long-term changes of JJA-specific humidity in the Arctic (see "Methods"). As shown in Fig. 3a, the long-term trend of JJA-specific humidity unrelated to AR activity has only increased in some subarctic areas, especially in the North Atlantic. The difference between this trend pattern (Fig. 3a) and the original one (Fig. 1b) is empirically attributed to long-term impacts of ARs in summer (Fig. 3b). More specifically, the moisture trends related to and unrelated to ARs in the Arctic (north of 60°N) are 0.017 g kg$^{-1}$ decade$^{-1}$ and 0.03 g kg$^{-1}$ decade$^{-1}$, respectively, indicating that the trend of ARs activity may contribute to 36% of the overall increase in atmospheric moisture across the Arctic since 1979. In addition, the impact of ARs is most pronounced in specific regions, including western Greenland, northern Europe, and eastern Siberia, where ARs account for 57.1%, 47.1%, and 67.8% of the corresponding increases in moisture over the period, respectively.

## Role of circulation variability in shaping AR activity in CESM1

The CMIP6 and CESM2 model response to historical radiative forcing indicates that the observed AR changes cannot be fully explained by anthropogenic forcing (Supplementary Fig. 5). Based on the statistical connection of ARs and circulation revealed by the aforementioned MCA analysis, we hypothesize that the large-scale circulation may direct more ARs into the Arctic region in recent decades. To further assess the plausibility of this linkage, we conduct three sets of simulations using CESM1. The first one is a CMIP6-type simulation with $CO_2$ forcing set at constant values of the year 2000 ($CO_2 = 370$ ppm) (referred to as "CTL" hereafter). This run aids in understanding how ARs behave in a climate state without changes in any radiative forcing. In the second set, in order to evaluate how long-term trends of JJA circulation exert a forcing on ARs in the Arctic, 3-D wind anomalies from the surface to the top of the atmosphere derived from long-term

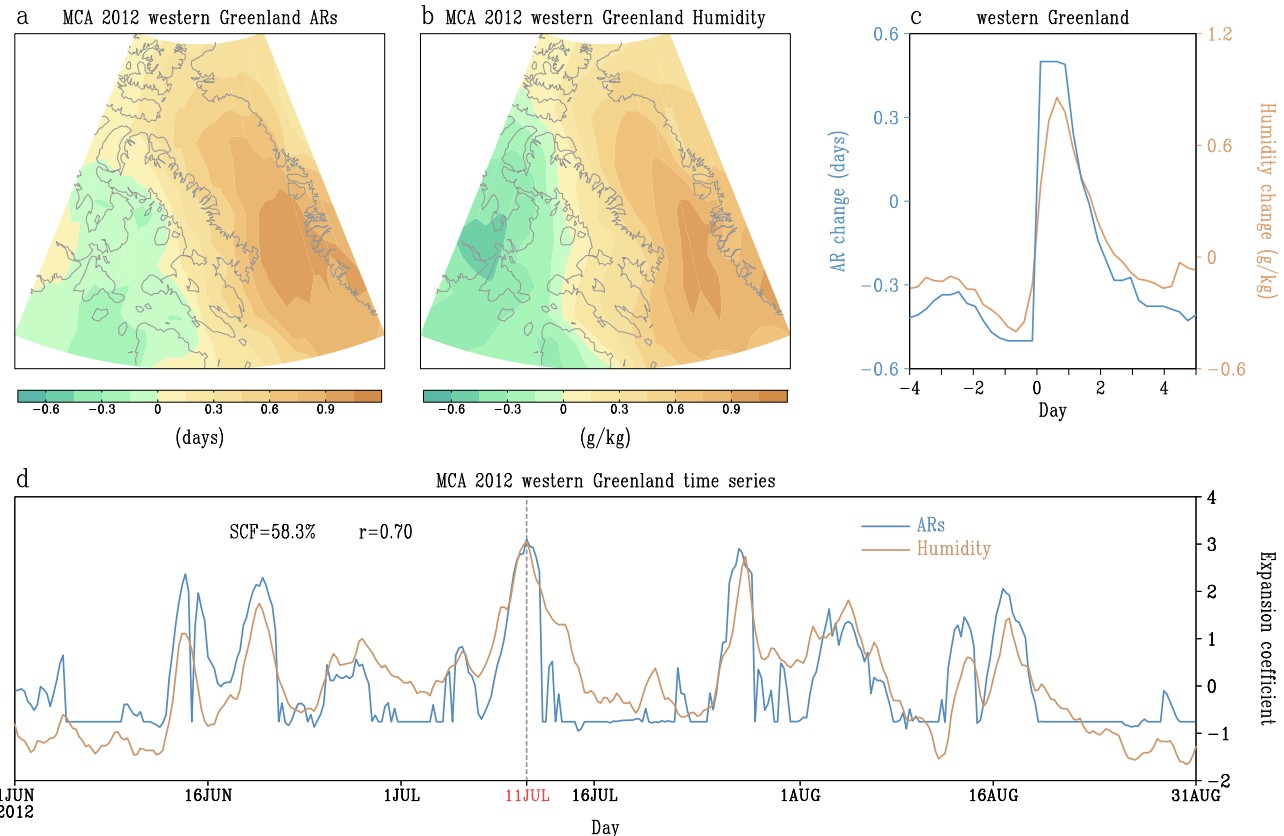

**Fig. 2 | Connections between atmospheric rivers (ARs) and specific humidity on a day-to-day time scale. a**, **b** Spatial pattern of the leading Maximum Covariance Analysis (MCA) mode of ARs (**a**, in days) and lower to middle tropospheric (surface to 500 hPa average) specific humidity (**b**, in g/kg) over western Greenland in summer 2012 (detrended and with climatological seasonal cycles removed). **c** Composite 6-hourly ARs (days) and lower to middle tropospheric specific humidity (g/kg) anomalies (from four days before to 5 days after the outbreak of ARs) within the western Greenland region from 1979 to 2019 (detrended and with climatological seasonal cycles removed). **d** Standardized time series of the leading MCA mode of ARs and lower to middle tropospheric specific humidity over western Greenland in summer 2012. "SCF" in (**d**) indicates the squared covariance fraction of the leading MCA mode. "r" in (**d**) indicates the correlation coefficient between the time series of the two patterns in the leading MCA mode. Numerous long horizontal blue lines in (**d**) indicate the absence of AR activity during the period. Both AR activity and humidity anomalies reached the maximum on July 11, which is marked in (**d**).

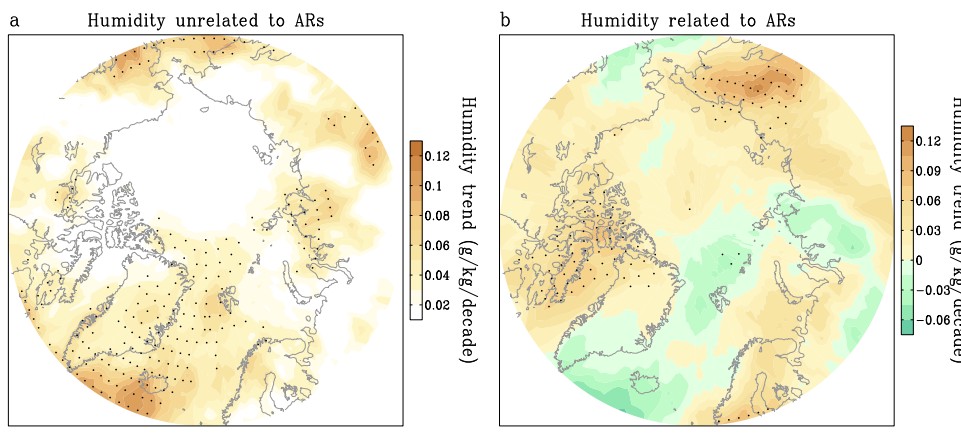

**Fig. 3 | Contribution of atmospheric rivers (ARs) to observed trends of specific humidity. a**, **b** Linear trends (g/kg/decade) of summer-specific humidity at lower to middle troposphere (surface to 500 hPa average) unrelated (**a**) and related (**b**) to the activity of ARs from 1979 to 2019. Black dots denote statistically significant trends at the 95% confidence level.

trends (m/s/decade) of JJA ERA5 circulation over the past 41 years (U and V) are added to the model during summer, only within the Arctic (north of 60°N) (see "Methods") (referred to as "WIN"). These anomalous 3-D winds (constant values in each grid cell over the 3 months) are incorporated in the model at each time step through the season, introducing seasonal mean wind anomalies without introducing any synoptic or intraseasonal variability in the wind fields. The goal of this design is to assess whether the model's response to these imposed wind trends can generate conditions conducive to ARs varying in a manner consistent with observations. The third set is the same as the second one except that anthropogenic forcing is imposed at the values of the year 2020 ($CO_2 = 415$ ppm) to include impacts of both

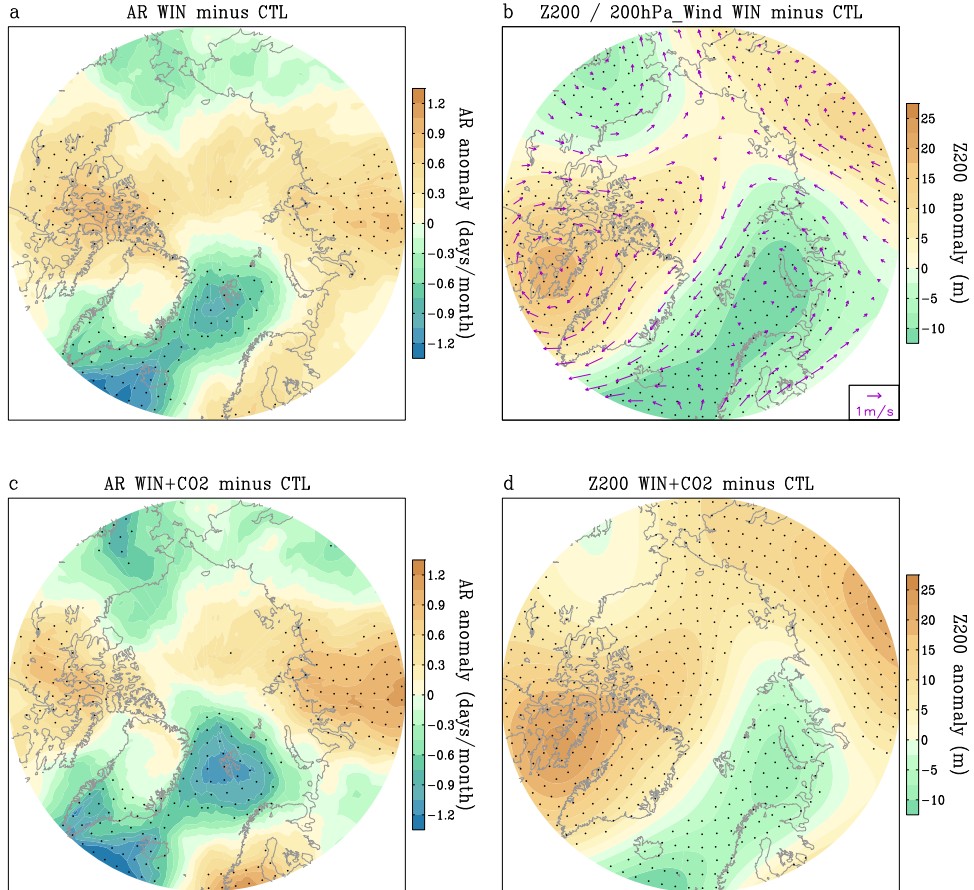

**Fig. 4 | Simulated influences of large-scale circulation and anthropogenic forcing on Atmospheric River (AR) changes. a, c** Differences of summer AR frequency (days/month) between the nudging run and the control run (WIN minus CTL) (**a**) and between the nudging+anthropogenic forcing run and the control run (WIN + CO$_2$ minus CTL) (**c**) based on the 40-year simulations. **b, d** Differences of summer geopotential height at 200 hPa (Z200) (m) between the nudging run and control run (**b**) and between the nudging+anthropogenic forcing run and the control run (**d**) based on the 40-year simulations. Wind anomalies (vectors, m/s) at 200 hPa in (**b**) are the differences of summer winds at 200 hPa between the nudging run and control run based on the 40-year simulations. The scale for winds is shown at the bottom-right corner of (**b**). Black dots denote statistically significant differences at the 95% confidence level for AR frequency in (**a, c**), and for Z200 in (**b, d**). WIN refers to the simulation with wind nudging used. CTL refers to the control run. WIN + CO$_2$ refers to the runs with both wind nudging and anthropogenic forcing imposed.

anthropogenic forcing and anomalous winds (referred to "WIN + CO$_2$"). The decadal change rate of CO$_2$ concentration from 2000 to 2020 ((415 ppm − 370 ppm) ÷ 2 decades ≈ 20 ppm/decade) is very similar to changes observed over the past four decades (-20 ppm/decade). This enables us to examine how anthropogenic forcing, along with the influence of the long-term wind trend pattern, regulated ARs from 1979 to 2019. Each set is continuously integrated for 40 years and the 40-year means of each set are used to estimate a relatively stable model response to the imposed forcing.

Figure 4a shows the difference of AR frequency between WIN and CTL simulations. It is clear that ARs occur more frequently over northwestern Greenland and the northern part of Eurasia, but less frequently in southern and eastern Greenland and the Bering Strait, similar to observations (Figs. 4a and 1a). The simulated ARs appear to be closely related to the large-scale circulation pattern. Southerly wind anomalies to the west of the high pressure over Greenland and central Siberia increase the frequency of ARs, while northerly wind anomalies to the east of the high pressure reduce AR activity (Fig. 4b, vectors). The positive AR response in northwestern Greenland and northern Europe is approximately 0.5 days per month, which is consistent with changes in observations (Figs. 4a and 1a). Around southern and eastern Greenland and the Bering Strait, the magnitude of the negative AR response is around −0.5 day per month (Fig. 4a). This implies that a wind field change in magnitude around 1 m/s can lead to significant JJA

AR changes. The simulated patterns of ARs and Z200 are consistent with the observed covariability of ARs and Z200 detected by MCA (Figs. 1e, f and 4a, b).

The difference in AR frequency and circulation between WIN + CO$_2$ and CTL is similar to that between WIN and CTL (Fig. 4a–d), suggesting that anthropogenic forcing in this model has a minor effect on AR activity in the Arctic. Since the response of ARs to anthropogenic forcing may be model dependent, we cannot exclude the possibility that some observed AR signals are indeed a consequence of global warming effects over the past few decades.

### Role of circulation variability in shaping AR activity using a fingerprint analysis

To further examine the impact of large-scale circulation on ARs we adapt a distinct approach, referred to as fingerprint analysis[63], based on the CESM2 large ensemble. In this approach, we use the AR and Z200 trends in the ensemble mean of 40 members to represent the forced response of the two fields to anthropogenic forcing, and the difference between each member with the ensemble mean as long-term trends due to internal variability. The ensemble mean of the ARs (Supplementary Fig. 5b) and Z200 trends (Supplementary Fig. 12) exhibit markedly different patterns from those in observations (Fig. 1a, c), echoing our findings derived from the CMIP6 result (Supplementary Fig. 5a), and reinforcing the notion that the observed AR

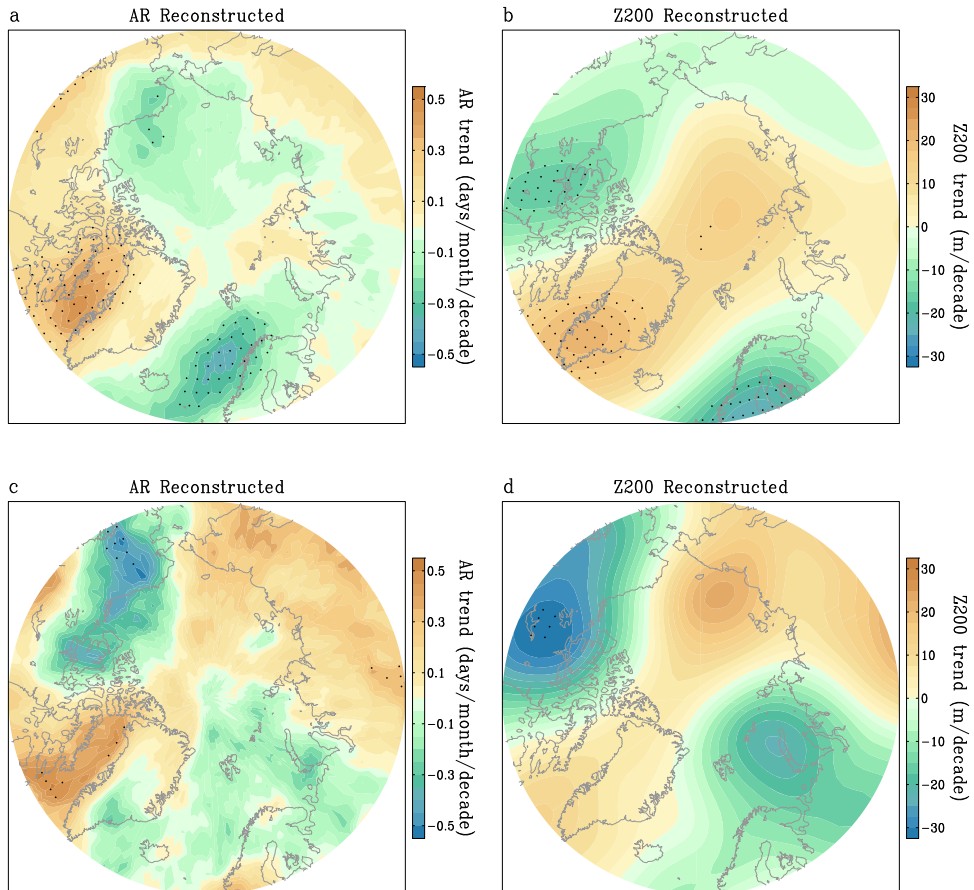

**Fig. 5 | Reconstructed trends of atmospheric rivers (ARs) and geopotential height at 200 hPa (Z200) in the subgroups of CESM2-LEN. a**, **c** Linear trends of summer AR frequency (days/month/decade) derived from the subgroup exhibiting strong increasing AR frequency trends over western Greenland (8 members) (**a**) and over both western Greenland and eastern Siberia (2 members) (**c**). **b**, **d** Linear trends in summer Z200 (m/decade) based on the subgroup with strong increasing AR frequency trends over western Greenland (**b**) and over both western Greenland and eastern Siberia (**d**). The 40 members from CESM2-LEN are used in this calculation (see "Methods"). Black dots denote statistically significant trends at the 95% confidence level.

and circulation trend patterns over the past decades cannot be fully explained by anthropogenic forcing.

To further understand how internal variability has contributed to the observed changes, we select a subset of 8 members from the 40-member ensemble, if their simulated AR frequency trends exhibit an increase (>80% percentile of simulated AR trends) over western Greenland. We then compare the large-scale circulation trend from this subgroup to examine whether the high-pressure pattern also emerges over Greenland, as observed. As we expect, the subgroup, possessing a better chance of replicating an increasing trend of ARs over western Greenland, also replicates the observed high-pressure trend over Greenland, indicating a strong constraint of large-scale JJA circulation trends on long-term changes of AR frequency over the region (Fig. 5a, b). Similar constraints of circulation on ARs are evident when focusing on northern Europe and eastern Siberia (Supplementary Fig. 13). Furthermore, two of the 40 members demonstrate an AR frequency increases trend in both western Greenland and eastern Siberia, and the corresponding large-scale circulation clearly exhibits a chain of high pressure in the above two areas as observed (Fig. 5c, d). This result suggests that long-term trends of large-scale circulation, particularly in forms similar to those observed, have a capacity to strongly modulate AR activity on low-frequency timescales. The establishment of this governing effect is attributed to a strong internal origin, as it is only discernible in a subset of the 40 members, rather than the ensemble means. This approach, emphasizing the internal variability of the model system, facilitates us to reasonably use model results to interpret observed features.

Our model experiments collectively support our initial hypothesis that the large-scale wind trend is able to apply a forcing to regulate changes of AR frequency, particularly over Greenland, northern Europe, and part of Eurasia. Over eastern Siberia, the wind-induced increase of AR frequency is shifted toward the interior of the continent in our nudging simulation results. However, the results obtained by the fingerprint method show that the large-scale circulation may still have some influences on ARs in that area (Supplementary Fig. 13c, d), indicating the need for further studies.

## Discussion

ARs are mainly driven by the atmosphere's baroclinic instability of the mean state and exhibit highly variable behavior on shorter timescales, but demonstrate sensitivity to low-frequency changes in the background flow. Over the past few decades, the summer season Arctic AR frequency has exhibited a well-organized increase in various regions, significantly influencing their local climates. To understand the underlying causes of these long-term changes in AR frequency, we employ a combination of observational and numerical modeling approaches. Our findings illustrate that a significant portion of the long-term AR frequency changes can be attributed to a steering flow effect generated by the large-scale circulation trend patterns, particularly characterized by a zonal wave number 2 structure in the Arctic. This circulation trend pattern involves a sequence of high- and low-pressure centers encircling the Arctic Ocean and Greenland, inducing corresponding shifts in AR activity along the same latitudinal band. By applying a statistical reconstruction method, we estimate that these

long-term trends in AR frequency have contributed to approximately 36% of the overall Pan-Arctic increase in atmospheric moisture and over ~50% of the increase in specific humidity over western Greenland, northern Europe, and eastern Siberia.

While the response of ARs to anthropogenic forcing in most climate models does not precisely match the observed pattern, we cannot rule out the possibility of indirect influences of anthropogenic forcing through changes in large-scale circulation and other systems, or the potential limitations of current models in capturing the forced response of ARs in a warming world. More in-depth analyses are needed, especially in regions close to eastern Siberia, where recent large-scale circulation trends appear to be of less significance in explaining long-term AR trends.

In addition, the Northern Hemisphere midlatitude jets are predicted to weaken and shift poleward due to anthropogenic influences in the coming decades[84]. How ARs in the Arctic may respond to a weaker, poleward-shifted jet stream and whether the identified impact of large-scale circulation on ARs, as shown in this study, will persist in a warming world alongside a more meandering jet remain open questions.

The increase in water vapor is a known contributor to enhanced cloud cover and increased downward longwave radiation in the Arctic, further exacerbating surface warming and the reduction of sea ice and land ice. These processes, collectively known as the moisture feedback, may also induce changes in large-scale circulation in the Arctic and beyond. However, the specific role of ARs, which are highly responsive to large-scale circulation variability over a wide range of timescales, in high-latitude moisture feedbacks remains unclear. In particular, ARs, which can cause rapid and extreme moisture surges, may lead to significant Arctic melt events and initiate positive feedback loops. Future studies are essential to explore these potential new roles of ARs in AA as the Arctic continues to warm.

## Methods
### Reanalysis and observational data
We examine changes in Arctic atmospheric circulation (Z200), specific humidity (For brevity, in some parts of the text, we use "humidity" and "specific humidity" interchangeably.), temperature, cloud cover, and radiation over the period from 1979 to 2019 using 6-hourly and monthly reanalysis data from ERA5[85] and ERA-Interim[86]. Our results are consistent between the two reanalysis datasets. Therefore, in this work, we focus on the findings derived from ERA5. All variables are regridded to a 1.5° × 1.5° resolution. We primarily utilize monthly reanalysis data to calculate the trend (calculated by the least-square regression method) of variables in our study (or for MCA). However, when identifying AR variability, 6-hourly data is ideal due to the transient and extreme nature of ARs, where higher temporal resolution data can better capture their temporal characteristics. Unfortunately, only a few CMIP6 models provide daily data (U, V, and specific humidity over multiple representative vertical levels) for the public, and 6-hourly data for these variables are not available. Consequently, we employ a remedial approach using daily data to detect ARs for these models. Monthly sea ice concentrations are acquired from NSIDC[87] for the analysis of the impact of ARs on sea ice variations.

### Large ensemble CESM2 simulations and CMIP6 experiments
We use 6-hourly data from 40 realizations of the CESM2 historical run[88] for the period spanning 1979-2014, each with slightly different initial atmospheric states. These 40-member CESM2 (CESM2-LEN) simulations are employed to explore the relationship between ARs and circulation and to assess how ARs respond to atmospheric internal variability. In addition, the multi-model ensemble mean of monthly output from the historical runs of 34 climate models archived in CMIP6[89] (Supplementary Table 1) is used to investigate the response of the atmosphere to historical anthropogenic forcing. We also incorporate daily data from the multi-model ensemble mean of 10 (as daily outputs needed for AR calculations are only accessible in these 10 members) climate models stored in CMIP6 historical runs (Supplementary Table 2) to calculate AR frequency and estimate the AR response to historical anthropogenic forcing. All of the CMIP6 model output mentioned above is focused on the period from 1979 to 2014. To facilitate comparison with the reanalysis data, we regrid all model datasets to a 1.5° × 1.5° resolution.

### Nudging experiments with the CESM1
To examine the regulation of large-scale circulation on AR activity in the Arctic in the fully coupled CESM1[73], we incorporate long-term trend wind patterns (the 200 hPa winds pattern is shown in Fig. 1c) from ERA5 to constrain the Arctic (north of 60°N) circulation in the WIN (with wind nudging only) and WIN + $CO_2$ (with wind nudging and anthropogenic forcing) runs from the surface to the top of the atmosphere. The magnitude of these wind trends is those changes spanning over 10 years (per decade). In each model step from June 1 to August 31 for these two runs (WIN and WIN + $CO_2$), within the nudging region, additional wind tendency terms, equal to the observed long-term wind trends, are added on top of the model's tendencies of zonal and meridional winds at each grid point. Since we impose constant wind anomalies each summer, this method acts to numerically incorporate observed wind trends into the model without introducing artificial short term variability. The same method has been used in a number of previous studies[37,70,71], which indicates the reliability of this approach in replaying observed circulation changes in the model. In addition, $CO_2$ concentration is held constant in CTL and WIN at the year 2000 level (370 ppm). In the WIN + $CO_2$ run, $CO_2$ is held constant at the value of the year 2020 (415 ppm). Other radiative and anthropogenic forcing, such as solar constant, other types of greenhouse gases, ozone concentration, various aerosols, land-use/land cover conditions are exactly the same between the two runs.

### AR detection
The ARs are defined via a detection algorithm based on integrated water vapor transport (IVT)[90]. In this algorithm, the 85th percentile of the climatological monthly IVT magnitude at each grid point is used as the threshold to identify ARs. IVT is calculated from ERA5 6-hourly wind and specific humidity fields integrated from 1000 to 300 hPa on 0.25° × 0.25° global grids. Additional criteria include the requirement that the length of the identified AR area should be greater than or equal to 2000 km and that the length-to-width ratio should be greater than or equal to 2. The ARs dataset is regridded to a 1.5° × 1.5° spatial resolution for analysis. This algorithm is recommended by the Atmospheric River Tracking Method Intercomparison Project (ARTMIP)[91], and has been extensively used in previous studies in detecting ARs in the Arctic and Antarctic[78,92–94]. In this study, ARs in the Arctic refer to all AR activity north of 60°N. Given that ARs represent transient and extreme moisture transport events, data with higher temporal resolution better describe their characteristics. Therefore, 6-hourly ERA5 reanalysis data are typically utilized to detect their signatures. We note that relatively few CMIP6 models (10 total) provide daily data as their highest temporal resolution data. As such, we only use this group to detect changes in ARs within the models. To study the year-to-year variability of ARs in the Arctic, a monthly mean AR field is created based on 6-hourly AR data. This field counts the number of AR activity in each grid point within each calendar month, with the value representing the monthly AR frequency (unit: days/month) in each grid point.

### AR frequency and specific humidity composites
To assess the impact of daily ARs on Arctic water vapor changes, we focus on strong AR events that occurred since 1979. These strong ARs are defined based on their continuous activity at the central points of

three key regions (western Greenland, northern Europe, and eastern Siberia) for a minimum duration of 24 h. It is important to note that the water vapor (specific humidity) anomalies associated with ARs are found to last behind the end of ARs by approximately 1 day. To ensure that the selected AR cases were not influenced by the prior water vapor changes resulting from preceding AR activity, we apply a criterion that requires these selected ARs to have no other AR activity on the day before their occurrence. Using this approach, we identify a total of 97 cases of ARs in the central point of western Greenland, 90 cases in northern Europe, and 83 cases in eastern Siberia. We then conduct composite analyses to examine the specific humidity anomalies associated with the AR events. Specifically, we analyze the changes in specific humidity (after detrending and removing the climatological seasonal cycles) for 4 days preceding and 5 days following the onset of ARs.

### Quantification of the contribution of ARs to specific humidity trends

We apply an AR removal method to isolate water vapor changes associated with 6-hourly AR activity. Initially, vertically integrated specific humidity fields for each summer (1979 to 2019) with 6-hourly resolution are generated. For each 6-hourly field from June 1st to August 31st, grid points with AR activity are determined by the "shape" of ARs as defined by the detection algorithm. When reconstructing the JJA seasonal mean humidity values in these grid points for any given summer, specific humidity values within these AR impact areas are excluded (with humidity over AR-shape grids set as zero). Conversely, averaging those 6-hourly grid points impacted by ARs (with humidity over non-AR grids set to zero) will form a JJA-specific humidity field related to ARs. This approach allowed us to reconstruct two JJA-specific humidity fields over the past 41 years, representing humidity variability unrelated to ARs and related to ARs, respectively.

### Description of the MCA approach

Maximum Covariance Analysis (MCA), often referred to as Singular Value Decomposition (SVD) analysis, is a valuable statistical method utilized in meteorological and oceanographic research to capture coupled modes of covariability between two fields[79]. Typically, this method operates on two distinct 2-D fields, X(m,t) and Y(n,t), where m and n represent spatial points and t represents the temporal dimension. MCA aims to compute a SVD of X×Y' (transpose matrix of Y), where each element in the matrix (X×Y') represents the covariance of a pair of points from X and Y, respectively (one spatial point from X and one spatial point from Y). This enables MCA to identify the dominant coupling modes between variations in X and Y. Specifically, we use MCA to extract the dominant coupled modes between the AR frequency and atmospheric variables over the Arctic in summer.

### Significance of correlation

The assumption for the statistical significance test of linear correlation is that the time series used to calculate the correlation consists of completely independent samples characterized by white-noise processes[95]. However, if the data contain strong autocorrelation (indicating some red-noise processes), this assumption may not hold well. To address this issue, it is necessary to consider the effect of effective sample size when assessing the significance of correlation by using conventional T test[96]. Accordingly, the statistical significance of the correlation coefficient is estimated by using an effective sample size $N^*$, which is given by:

$$N^* = N \frac{1 - r_1 r_2}{1 + r_1 r_2} \tag{1}$$

where $N$ is the number of available time steps, $r_1$ and $r_2$ are lag-one autocorrelation coefficients of each variable[97].

## Data availability

ECMWF ERA5 reanalysis product is available at https://www.ecmwf.int/en/forecasts/datasets. Simulated circulation, specific humidity, and temperature under anthropogenic forcing were obtained from CMIP6 and CESM2 archive at https://esgf.llnl.gov/ and IBS openDAP server. Sea Ice Concentrations from Nimbus-7 SMMR and DMSP SSM/I-SSMIS Passive Microwave Data, Version 2 are accessed from NSIDC at https://nsidc.org/data/nsidc-0051/versions/2. The AR data and CESM nudging experiment output are available from the corresponding authors upon request. The data used to make the plots for this paper can be accessed at https://zenodo.org/records/11159190.

## Code availability

All code necessary for performing the reported analyses is available upon request from the corresponding authors.

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

## Acknowledgements

This study is supported by NSF Arctic System Science awards OPP 2246601 and OPP 2246600, and Climate Variability & Predictability (NA23OAR4310273) programs as part of NOAA's Climate Program Office. Z.W. acknowledges support from the National Natural Science Foundation of China Grants 42105028, the China Postdoctoral Science Foundation 2020T130640, the Natural Science Foundation of Hunan Province of China 2024JJ6101, and the Open Grants of the State Key Laboratory of Severe Weather 2022LASW-B23. G.H. acknowledges support from the National Natural Science Foundation of China 42141019 and 41831175. R.W. received support from the National Natural Science Foundation of China Grants 41721004.

## Author contributions

Q.D. and T.B. conceived the study. Z.W. analyzed the data and created the figures. Q.D. and Z.W. led analyses, interpreted results, and wrote the paper. Q.D. created the nudging simulations and designed the fingerprint analysis. T.B. and R.W. provided insightful comments. G.H., W.C., B.G., S.C., X.C., Z.C., D.B., D.N., I.B., D.T., and Z.L. contributed to discussions and provided comments.

## Competing interests

The authors declare no competing interests.

## Additional information

Qinghua Ding or Thomas J. Ballinger.

