## [Peer Review File · Nature Communications]

Role of atmospheric rivers in shaping long term Arctic moisture variabilityREVIEWER COMMENTS

Reviewer #1 (Remarks to the Author):

This manuscript explores the trend in atmospheric rivers in the Arctic in boreal summertime and the drivers of this change. Using observations and models, including experiments with winds nudged to observational trends, they found that large-scale circulation trends played a key role in atmospheric rivers in the Arctic and that they contribute to the increased wetting trend in the Arctic summer. Understanding the variability and long term trends of humidity in the Arctic is vitally important in terms of changes precipitation, temperature, sea ice etc and so an analysis of this is necessary. However this paper at times seems a little disjointed, jumping between indicating that humidity is a driver of AR to AR driving humidity or that the paper is about the trends of humidity. I think clarifying the message of the paper better is necessary. Furthermore I feel that there should have been more analysis done on other sources of humidity in the Arctic or simply remove those completely and focus only on AR related humidity in summer time. While I find the model experiments with nudging the winds in the model interesting, more work is required to make the manuscript publishable. I have outlined a number of concerns/questions about this below.

Majors

I am curious about the role of other sources of moisture in the Arctic namely as Figure 3a shows that there is a large significant portion of the Arctic in summer that is experiencing increasing humidity that is not related to ARs and this should be investigated more thoroughly or have more separation between AR induced humidity relative to influence of remote sources (sea ice loss) for example.

Model bias was never discussed at all which I think is important. If the observations are showing that we are getting an increase of moisture flux from ARs through changes in circulation patterns, but the models do not show this, then an exploration into the wind/circulation trends in models relative to observations would be a first step to understanding the differences between the models and the observations rather than stating it is solely due internal variability.

This is all assessing the trends from 1979-2014 but I feel a brief analysis of the climatology is needed to assess what these trends are in relation to particularly for the circulation, is more negative trend that it is more negative or a shift in the centres of action? for example AR trends show a decrease trend over Alaska were in the mean this region has the most AR's climatologically. I think it would be worth also referencing this within your study, is the circulation changes contributing to this negative trend in ARs in that location?

Details

Line 82: You state that summer months get more ARs because the jet is further poleward but further poleward relative to what. Supp fig 1b just shows climatological winds and AR so not evident that it is more poleward e.g. is this relative to DJF – then maybe say or show that too.

Line 91: What are the thermodynamic effects from global warming the ARs are sensitive to?

Line 95: Are the “recent observed features” the increase in ARs?

Line 107: Is there any references/papers to support this claim that models lack the ability to capture the observed circulation patterns in the Arctic?

Line 110: Why did you decide to go for wind rather than geopotential height for example for the circulation trends?

Line 120: Counting the number of AR activity by grid – what do you mean by grid here, is it West Greenland or by gridcell as this is what is usually thought when one refers to ‘grids’ in a modelling paper. This lack of distinction in present throughout the manuscript particularly in the methods (e.g. lines 398 and 422) and is confusing so I suggest changing the name e.g. region.

Line 133: atmospheric temperature is fig 1d not 1c as stated here.

Line 143: Relating to the uniform rise in z200 over Greenland and Eurasia, I would not say that this is uniform, its much greater in Europe than Greenland, and with the white colour even make it appear, at first glance to have no trend. This could be written better to address this slight differences which are likely from internal variability.

Line 147-149: I am not sure how storm track shifts over the North Pacific may have impacted increasing trends of AR, has this been documented somewhere before? Have you does analysis to show this?

Line 165-168: I find this difficult to read, you say that “Z200-AR may share common mechanisms” which to me suggests that something else controls BOTH Z200 and AR but isn’t it that the circulation aka Z200 that drives AR? And the second half ‘suggesting a deeper understanding” is very dense to read.

Line 170: The text says tropospheric winds but figure 1f says 200hPa winds so have you assessed winds throughout the vertical column or just at the top of the troposphere? If latter state that this is the upper level westerly winds and remove the suggestion that this is throughout the whole troposphere.

Line 174: Can you explain how this shows interannual time scales if it is days/month, is it also per year?

Line 188-193: Have you shown that AR are statistically connected to humidity and temperature, figure 1 a-d shows only the linear trends but does not link them statistically? And was there ever a suggestion previous to this line that humidity and temperature drove AR trends, especially when in lines 192 you state the humidity is associated with AR propagation into the Arctic indicating that AR therefore is a driver of humidity. Revise this section and be very specific by what you mean.

Line 204: How many ARs were there in 2012 over Greenland?

Line 213-214: As seen in supplementary figure 7.

Line 222-224: This is a little cryptic, what other regions are these? Are there figures for them?

Line 230-233: This commentary is confusing to me, you say that figures 3a and 1b are similar (I agree here) but that this is similar to figure 3b, which I do not see. Figures 3a and 1b show increases of humidity everywhere while humidity related to ARs (figure 3b) shows negative trends so can you explain this a little more.

Furthermore, is a sum of Figs 3a and 3b result in fig1b – meaning that you get increase trend of humidity everywhere.

Line 241: I do not understand this, you don't actually look at AR changes in CMIP6 models but just humidity, Z200 and temperature?

Line 270: Are you able to actually show the wind anomalies in Figure 4b rather than infer based on the Z200 circulation?

Lines 287-293: Have you plots made to show this? If so, reference them as I am not sure what you are talking about/showing here.

Line 295: What is the size of the subgroup?

Line 303: You state here that you only use 2 of the 40 members but don't state this in the figure caption so just want to check what you are actually showing in figure 5c and d.

Line 309: You have shown previously that the models do not do a good job replicating the trends of AR and Z200 and yet here are saying that they agree with your results and so I am confused about this?

Figures

Supp fig 5 and 8: In my version of documents, these plots are cut off and so difficult to attest to claims made about these figures in your manuscript.

Methods:

Line 389: Be specific here, what are the anthropogenic forcings that are held constant?

Line 416: I would just note that you are examining the specific humidity anomalies here, even though it is mentioned in following line, will avoid any confusion about what is meant.

Much more detail is needed about the nudged model specifically demonstrating what the nudged field looks like.

Significance of correlation, you never explain why it is important to account for autocorrelation in the time series

Reviewer #2 (Remarks to the Author):

Review of the manuscript NCOMMS-23-52552 entitled “Role of atmospheric rivers in shaping long-term Arctic moisture variability” by Wang et al.

The manuscript shows the influence of the large-scale atmospheric circulation in modulating the atmospheric rivers moistening impact on the Arctic using both the ERA5 reanalysis and idealised simulations with a climate model, in order to explain the trends in the observed atmospheric rivers frequency. The paper is relatively well written. However, I find that many aspects (methods and results) could be better explained, especially the maximum covariance analysis, on which I am not an expert myself. Therefore, I recommend major revisions of the manuscript.

Here below, are my comments listed in the order of the manuscript followed by a few technicalities.

Fig. S1a and line 82: The figure represents a mean frequency of atmospheric rivers (AR frequency), not a mean. Please modify the plot and the caption. Moreover, it is not clear how this AR frequency in the Arctic is defined. Is it the mean of the AR frequency north of 60N? Is it the mean of AR frequency at 60N? In this latter case, are only the poleward ARs considered or are all ARs considered? Please explain better in the caption as well as in the Methods section.

Fig. S1b and lines 82-84: To show that the AR frequency and the jet are shifted northward in summer compared to winter, the authors should have represented the difference between winter and summer and not summer only. This behaviour of the jet is well known, but it maybe less known for the ARs. Please consider adding this plot (difference or superimposition of the winter and summer fields in contours).

Fig. S2: In the caption, (blue fan-shaped covered area, → (blue fan-shaped covered area)

Lines 115-117: the authors mention “moistening trends”, but they do not show any specific humidity trends, for example. It seems that they assimilate this wording to the AR frequency trend. However, it is not fully correct as ARs are not the only source of humidity in the Arctic. The authors also wrote “the Pan-Arctic moistening trend is the most positive and significant (Fig. S1a)”, but they do not show any significance calculation and they should rather refer to Fig. S2b, which shows a trend, shouldn't they? Ideally, the authors should rather represent the specific humidity trend in summer and winter.

Line 130: The authors should also add Western Alaska here because it seems to be one of the most prominent pathways of AR into the Arctic in summer, with the largest AR mean frequency north of 60N as seen in Fig. S2a.

Line 133: Why using the geopotential at 200 hPa? It is in the stratosphere at those high latitudes. Could the authors explain why they use this pressure level and not at a lower level, such as 400 or 500 hPa?

Line 133: Fig. 1b-1c → Fig. 1b-d (to including panels b, c, and d)

Line 136: The authors should describe the methodology of the maximum covariance analysis in the Methods section and what results it will give so that readers not familiar with this method can understand it.

Lines 130-137 could be written more concisely. Consider rewriting.

Lines 138-144: What about adding SSP5-8.5 to complete the historical simulations and calculate the trend over the exact same period as in the reanalysis (same as what has been performed for CESM2 in Fig. S4b it seems)? Or what about using only 1979-2014 in the reanalysis to match the CMIP6 historical period? A few more or less years can make a difference when calculating trends (see for example Fig. 2 of Outten et al. 2022 <https://wcd.copernicus.org/articles/4/95/2023/wcd-4-95-2023.pdf>).

Line 142-143: why not using the same vertical extent for all variables? Moreover, these vertical extents are different than the one used to calculate the IVT (line 399). Can the authors justify their choice?

Line 144: Could the authors add a measure of significance on Fig. S3? An easy one could be the number of models agreeing on the sign of the trend. Moreover, are there some CMIP6 models in better agreement with ERA5 than others? Could the answer to this question be mentioned in the manuscript?

Fig. S4a: Why is this figure the mean of only 10 CMIP6 models when Fig. S3 includes 40 models? Could Fig. S3 be redrawn with the 10 same models?

In the caption: “and 40-member” → and (b) 40-member

Line 148: What are the SST changes the authors mention here? Please be more precise. Why would it be the only factor influencing the AR trend found in CMIP6?

Line 165: What are “the long term changes of ARs and circulation”? Do the authors mean the trend?

Line 181: Shouldn't it be MCA1 ARs-Z200 instead of MCA1-ARs or MCA1-Z200 since the maximum covariance analysis is performed using both the AR frequency and Z200? Do the authors expect different results by using MCA1-ARs instead of MCA1-Z200? Please clarify the MCA methodology.

Line 195: Greenland → the Western Greenland region

Lines 196-198: Could the authors add a reference to this statement?

Line 199: What do the authors mean with “daily AR frequency”? Aren’t the daily fields binary masks (0-1 fields) of atmospheric rivers?

Why do the authors use both CESM1 and CESM2 in their study? Would it be possible to use only one model version so that the results of their two analysis are comparable? Do the authors expect the two versions of CESM to give similar results for their idealised experiments? This uncertainty could be acknowledged somewhere in the manuscript.

Lines 206-207: Isn’t that result coming directly from the MCA? Would the authors expect another result? Also, the figure referred to should be Fig. 2d not 2c, right?

Figure 2d: - Could the authors give more details on what is represented on panel d? Why is the y-axis label saying “Standard deviation” whereas the caption text says “standardized time series”? Please clarify.

- Moreover, the straight horizontal parts in the blue curve highlight the days with no AR. Is that right? Isn’t it an issue to perform the MCA with some time series having 0s?

- Finally, it is not clear to me if this time series (panel d) is for the central point of the region or an average over the region or something else. Please add this precision.

Lines 207-208: Again, what does the value of the time series peak on 11 July (Fig. 2d) mean? I understand that the fact above 0 values mean that there is an atmospheric river and that many days around the 11th experienced an atmospheric river. Is that correct?

Line 209: remove).

Figure 2c and Figure S7: Why do the two figures have different y-axis label (AR change (days) for Fig. 2c and AR anomaly (case) for Fig. S7). Please adjust and make it consistent (both the label and the unit).

Lines 228-230: The methodology is not clear at all to me. I suggest a major rewriting of the method section for this analysis.

Lines 233-234: How are those trends calculated? Please explain in the text.

Figure 3: My understanding is that Fig. 3b is obtained by subtracting Fig. 3a from Fig. 1b. Is that correct? In Eastern Siberia, it looks like the sum of Fig. 3a and b would be higher than Fig. 1b. Could the authors check? In any case, please add some explanations to the methods (as also written above).

Lines 241-242: I am sorry, but I do not understand how the authors can say that from Fig. S3a. Could the authors elaborate a bit here?

Line 254: Does the addition of the anomalies introduce noise or discontinuities in the wind field?

Line 276: How small is this wind change?

Line 291: "...and again, and reinforcing..." needs rewriting.

Line 366: add (Z200) after "Arctic atmospheric circulation"

Lines 367-368: Why do the authors use both 6-hourly and monthly reanalysis data? Moreover, since data from models are available daily, why not also use daily reanalysis, instead of 6-hourly? It looks inconsistent, especially as the ARs are detected from daily fields for the models but 6-hourly fields for the reanalysis.

Line 398 (and others, e.g. 425, 426): "grid" → grid point

Following the Oxford dictionary, the noun grid means "An arrangement of parallel bars with openings between them". Therefore, the authors seem to misuse this noun. Please correct where appropriate.

Line 410: "lag" or last?

Lines 422-423: "Initially ... algorithm". This sentence is not understandable to me. Please adjust and clarify.

Line 424: "extended spatiotemporally". The spatial extension is of 6 degrees, but how long in time is the AR impact extended?

Line 426: What do the authors mean with "time point"?

Lines 428-429: How can the AR impacts be removed? Please add an explanation.

Figures:

- Please represent the coastlines in black. They are barely visible in gray.
- Figure 1 would benefit a lot from larger panels with for example only 3 panels on the top row.
- How is the significance calculated (the black dots in the figures)?

Technicalities (probably not exhaustive):

Line 53: altered Arctic → altered the Arctic

Line 78: move eastward → move (north)eastward

Line 135: in the Fig. S2b, Fig. 1c → in Fig. S2b, Fig. 1c

Line 399: $0.25^\circ \times .25^\circ$ → $0.25^\circ \times 0.25^\circ$

Line 442: is → are

Line 655: The dotted → Black dots

Lines 681, 690, 699: The dotted grids → Black dots

RESPONSE TO REVIEWERS

We express our gratitude to all the reviewers for dedicating their time to evaluate our work, providing both positive feedback and constructive criticism on our manuscript. Their insightful comments and suggestions have significantly contributed to enhancing the presentation and clarity of our key arguments. We have fully integrated the reviewers' recommendations into the revised manuscript, encompassing the following three key changes outlined below.

1. To address the concerns raised by the reviewers, we have made several revisions to the text. In the introduction, we have rewritten various sections to discuss multiple sources contributing to the recent moistening trend in the Arctic (Lines 58-62) and have provided explanations for why ARs are sensitive to thermodynamic effects of global warming (Lines 103-106). Furthermore, we have emphasized the connection between extratropical storm tracks and ARs (Lines 84-88). In the Methods section, we have clarified why data with different temporal resolutions were utilized (Lines 413-418) and elucidated the methodology for distinguishing moisture variability related to ARs from that unrelated to ARs (Lines 482-495). Additionally, we have introduced a new segment in the Methods to elucidate the fundamental principles of the MCA method and have modified the discussion on the significance test of correlation in response to the reviewer's concern.
2. Some parts of the main text and supplementary materials have been reorganized, and numerous plots have been redrawn to streamline our presentation and improve clarity of reasoning.
3. Additional analyses have been conducted, and if the results are not shown in the text, they are provided for the reviewers' reference along with our response to their inquiries.

Below, you will find our comprehensive responses (highlighted in blue font) to the reviewers' comments. We hope that we have effectively addressed all concerns raised in the revised manuscript.

REVIEWER COMMENTS

Reviewer #1 (Remarks to the Author):

This manuscript explores the trend in atmospheric rivers in the Arctic in boreal summertime and the drivers of this change. Using observations and models, including experiments with winds nudged to observational trends, they found that large-scale circulation trends played a key role in atmospheric rivers in the Arctic and that they contribute to the increased wetting trend in the Arctic summer. Understanding the variability and long term trends of humidity in the Arctic is vitally important in terms of changes precipitation, temperature, sea ice etc and so an analysis of this is necessary. However this paper at times seems a little disjointed, jumping between indicating that humidity is a driver of AR to AR driving humidity or that the

paper is about the trends of humidity. I think clarifying the message of the paper better is necessary. Furthermore I feel that there should have been more analysis done on other sources of humidity in the Arctic or simply remove those completely and focus only on AR related humidity in summer time. While I find the model experiments with nudging the winds in the model interesting, more work is required to make the manuscript publishable.

I have outlined a number of concerns/questions about this below.

We extend our gratitude to the reviewer for his/her positive remarks and valuable suggestions, which have significantly improved the clarity and coherence of our manuscript. We hope that the reviewer will find the modifications we have made satisfactory in response to the comments and suggestions.

Majors

I am curious about the role of other sources of moisture in the Arctic namely as Figure 3a shows that there is a large significant portion of the Arctic in summer that is experiencing increasing humidity that is not related to ARs and this should be investigated more thoroughly or have more separation between AR induced humidity relative to influence of remote sources (sea ice loss) for example.

Thank you for your constructive comment. Indeed, the increasing moisture over the Arctic in recent decades has been attributed to multiple sources, as documented in previous studies. These sources include local ocean and land surface evaporation, sublimation of ice and snow, as well as transport from lower latitudes (Bintanja and Selten, 2014; Vihma et al., 2015). It is widely believed that this increase is primarily due to the warming induced by anthropogenic forcing, which leads to higher moisture levels in the atmosphere and contributes to climate change in the Arctic (Serreze et al., 2012; Boisvert and Stroeve, 2015; Olonscheck et al., 2019; Bokuchava and Semenov, 2021; Choi et al., 2023). However, what has received less attention is how changes in circulation patterns regulate moisture transport, particularly through alterations in AR characteristics, which is the main focus of this study.

We have incorporated an additional sentence to offer a brief review on other sources of moisture in the Arctic as listed above in lines 58-62 of the manuscript:

“This increase in moisture is prominent throughout the year, with the most significant rise occurring during the summer months (June-July-August, abbreviated as JJA)^{14,15}, which has been attributed to multiple sources triggered by global warming, including increased evaporation from local ocean and surrounding continents¹⁶, enhanced sublimation of ice and snow within the Arctic^{17,18}, and intensified moisture transport from lower latitudes^{19,14,20,21,22,23,24,25,26}.”

Model bias was never discussed at all which I think is important. If the observations are showing that we are getting an increase of moisture flux from ARs through changes in

circulation patterns, but the models do not show this, then an exploration into the wind/circulation trends in models relative to observations would be a first step to understanding the differences between the models and the observations rather than stating it is solely due internal variability.

This aspect has been addressed in a series of previous studies, including published work from our group (Ding et al., 2017; Baxter and Ding, 2022; Li et al., 2022; Topál et al. 2022, 2023; Feng et al. 2023), and by other colleagues (Athanasé et al., 2022; Roach and Blanchard, 2022). In brief, due to the model's limitations in simulating the mean state, tropical-Arctic teleconnections (Feng et al., 2023), and certain low-frequency SST modes, the models generally exhibit weaknesses in capturing observed circulation variability in the Arctic. By nudging observed wind trends within a model (Ding et al., 2022, Roach and Blanchard, 2022), it becomes feasible to accurately simulate the response of local climate fields, such as summer sea ice melting. Furthermore, utilizing a fingerprint analysis (emphasizing both internal and forced components of model simulations), we can achieve a better match in simulating observed changes in Arctic climate over the past decades. Additional clarification has been integrated to elaborate on this point further (Lines 164-167).

This is all assessing the trends from 1979-2014 but I feel a brief analysis of the climatology is needed to assess what these trends are in relation to particularly for the circulation, is more negative trend that it is more negative or a shift in the centres of action? for example AR trends show a decrease trend over Alaska were in the mean this region has the most AR's climatologically. I think it would be worth also referencing this within your study, is the circulation changes contributing to this negative trend in ARs in that location?

Thank you for your suggestion. In response, we have revised Supplementary Fig. 3 to include the summer climatology of four variables (1000-500 hPa specific humidity, 200 hPa geopotential height and winds, and 1000-200 hPa air temperature) in the Arctic from 1979 to 2019, and we have incorporated relevant discussions in the manuscript (Lines 141-143).

As suggested, we emphasize the negative trend in AR activity over Alaska, which is an area of the Arctic where these features are climatologically prevalent. Our analysis indicates that circulation patterns may also contribute to this trend in Alaska (Lines 151, 155-156).

Details

Line 82: You state that summer months get more ARs because the jet is further poleward but further poleward relative to what. Supp fig 1b just shows climatological winds and AR so not evident that it is more poleward e.g. is this relative to DJF – then maybe say or show that too.

Thank you for your suggestion. What we meant is that during summer, the jet stream is positioned further north compared to its location in other seasons. To clarify, we have included the climatological ARs and u200 for DJF in Supplementary Fig. 1 as panel c. Additionally, we have adjusted the sentence accordingly (Lines 92-94, Fig S1c).

Line 91: What are the thermodynamic effects from global warming the ARs are sensitive to?

The formation and action of ARs rely on adequate kinetic energy and abundant moisture. With global warming, the available potential energy (a source of baroclinic instability) and atmospheric moisture are expected to increase, consequently enhancing the mean occurrence of ARs and leading to long term variability in their frequency, intensity, trajectory, and duration characteristics. However, this is a complex area of study. Recent research suggests that the most intense midlatitude storms will shift poleward due to tropical expansion and increased atmospheric instability (Tamarin-Brodsky and Kaspi, 2017; Yang et al., 2020; Gentile et al., 2023). Whether this poleward shift will extend to the Arctic remains an open question, and we have provided a better summary of this point in the introduction (Lines 103-106).

Line 95: Are the “recent observed features” the increase in ARs?

It refers to both ARs and the moistening trend in the Arctic. We have rephrased this part to ensure clarity.

Line 107: Is there any references/papers to support this claim that models lack the ability to capture the observed circulation patterns in the Arctic?

Supplementary Fig. 3b supports this point. We have cited a number of references to support this claim in the revision (Line 121).

Line 110: Why did you decide to go for wind rather than geopotential height for example for the circulation trends?

Geopotential height changes reflect both temperature and wind field variations, as these two variables are closely interlinked physically. However, the main distinction between the two fields lies in the fact that temperature changes can directly result from both external and internal factors, whereas wind patterns, particularly regional wind patterns, are primarily influenced by pressure changes. In a first-order approximation, anthropogenic forcing does not significantly alter the horizontal distribution of air pressure. Therefore, changes in wind patterns are less sensitive to anthropogenic forcing. This is evident in many models' responses to anthropogenic forcing, which often prioritize long-term trends in temperature over changes in wind patterns. Given that the response of ARs in most models' forced simulations does not closely resemble observed changes, we speculate that observed AR changes could be internally driven. To further examine this idea, our primary focus is on examining the influence of wind patterns on ARs.

Line 120: Counting the number of AR activity by grid – what do you mean by grid here, is it West Greenland or by grid cell as this is what is usually thought when one refers to ‘grids’ in a modelling paper. This lack of distinction in present throughout the manuscript particularly in the methods (e.g. lines 398 and 422) and is confusing so I suggest changing the name e.g.

region.

Thank you for your suggestion. The AR frequency data is derived from the ERA5 reanalysis, which has a resolution of $1.5^{\circ} \times 1.5^{\circ}$, resulting in a total of 5040 (21×240) grid points covering the Arctic region. To assess the frequency of AR occurrence across the Arctic, the mean AR activity (from all 5040 grid points) within the region was calculated. Therefore, the sentence should read as 'this field counts the number of AR occurrences at each grid point'. We have also made wording adjustments to address several other similar issues (new lines 136, 454, 484).

Line 133: atmospheric temperature is fig 1d not 1c as stated here.

We modified the sentence, so it only refers to geopotential height in the revised manuscript.

Line 143: Relating to the uniform rise in z200 over Greenland and Eurasia, I would not say that this is uniform, its much greater in Europe than Greenland, and with the white colour even make it appear, at first glance to have no trend. This could be written better to address this slight differences which are likely from internal variability.

Thank you for your suggestion. You are right that the trend of 200 hPa geopotential height (Z200) is relatively weak over Greenland, but there are noticeable and relatively uniform increases of Z200 over Europe. We have revised the sentence to provide a more accurate description (Lines 160-164).

Line 147-149: I am not sure how storm track shifts over the North Pacific may have impacted increasing trends of AR, has this been documented somewhere before? Have you does analysis to show this?

Storm tracks, recognized as regions of intense synoptic eddy activity, are closely tied to extratropical cyclone and anticyclone activity (Blackmon 1976). ARs form as a result of the interaction between a cold front, which transports a substantial amount of water vapor into the warm sector, a warm front (Dacre et al. 2015). Thus, ARs represent a unique and extreme phenomenon linked to the extratropical storm tracks (Lavers et al. 2020). We have added sentences and citations to relevant literature to support this explanation (Lines 84-88).

Line 165-168: I find this difficult to read, you say that “Z200-AR may share common mechanisms” which to me suggests that something else controls BOTH Z200 and AR but isn't it that the circulation aka Z200 that drives AR? And the second half ‘suggesting a deeper understanding’ is very dense to read.

The sentence is too long, which may cause some confusion. We modified it and showed it in the last sentence of the previous paragraph as follows: “This resemblance suggests that the secular trends of Z200 and ARs over Greenland may physically connected, as indicated by their interannual connections identified by MCA1.” We hope our edit improves the

readability of this sentence (Lines 190-192).

Line 170: The text says tropospheric winds but figure 1f says 200hPa winds so have you assessed winds throughout the vertical column or just at the top of the troposphere? If latter state that this is the upper level westerly winds and remove the suggestion that this is throughout the whole troposphere.

The sentence refers to the summertime climatological westerly winds that prevail in the Arctic troposphere. We have added a figure in supplementary material (Supplementary Fig. 3c, vectors) and a few words to clarify the sentence (Line 198).

Line 174: Can you explain how this shows interannual time scales if it is days/month, is it also per year?

The unit of ARs frequency is days per month, indicating the number of days in a month when ARs occur at a grid point. When we calculate the JJA mean, we average the values from June, July, and August (JJA) for each year, and then compare year-to-year AR changes.

Line 188-193: Have you shown that AR are statistically connected to humidity and temperature, figure 1 a-d shows only the linear trends but does not link them statistically? And was there ever a suggestion previous to this line that humidity and temperature drove AR trends, especially when in lines 192 you state the humidity is associated with AR propagation into the Arctic indicating that AR therefore is a driver of humidity. Revise this section and be very specific by what you mean.

The relationship between ARs and humidity is evident across various time scales. Figure 1 illustrates the long-term trend relationship of variables, while Figures 2a, 2b, 2d, S7, S8, and S9 depict the day-to-day relationship between ARs and moisture. To further elucidate the statistical connection between ARs and humidity, we composited individual AR events and corresponding specific humidity in specific regions (western Greenland, northern Europe, eastern Siberia, near the North Pole, and southwestern Alaska, Figs. 2c, S10, and S11, respectively) in the subsequent part of this section (lines 245-256). A series of MCA analyses is done to further show a close relationship between ARs and specific humidity and ARs with temperature on year to year time scales in the Arctic (Figs. Rsp1 and Rsp2, which were not shown in the main text). These figures clearly demonstrate the relationships among ARs, specific humidity, and air temperature.

Figure Rsp1. a-b, Spatial pattern of the leading MCA mode of summer ARs (**a**, in days/month) and specific humidity (**b**, in g/kg) (detrended). **c**, Standardized time series of the leading MCA mode of summer ARs (blue line) and specific humidity (red line). ‘SCF’ in **c** indicates the squared covariance fraction of the leading MCA mode between summer ARs and specific humidity. ‘r’ in **c** indicates the correlation coefficients between the time series of the leading MCA mode (summer ARs and specific humidity).

Figure Rsp2. Same as Fig. Rsp1, but for the variables between ARs and temperature.

Line 204: How many ARs were there in 2012 over Greenland?

We have observed a total of 29 AR events that reached western Greenland during the summer of 2012. Specifically, there were 7 events in June, 13 in July, and 9 in August. It's notable that the number of AR events in the summer of 2012 significantly surpasses the summer climatology (average 2-3 events in a month) of ARs from 1979 to 2019 (Supplementary Fig. 1a-1b)

Line 213-214: As seen in supplementary figure 7.

Supplementary Fig. 7 supplements Figure 2c in the text, but extends the period to 1979-2019. Previously, we did not include the MCA for the regions over northern Europe and eastern Siberia in the original supplementary material. In the revision, we have added these two figures to the supplementary material as Supplementary Fig. 7 and Supplementary Fig. 8, respectively, and reorganized the numbering of supplementary figures accordingly.

Line 222-224: This is a little cryptic, what other regions are these? Are there figures for them?

The regions showing notable increase trends in AR frequency, are evident near the North Pole,

while a significant reduction in ARs is observed on southwestern Alaska. We present the corresponding figure below (Fig. Rsp3, Supplementary Fig. 11). We modified the sentences of the specific figure (Lines 252-256).

Figure Rsp3. a-b, Composite 6-hourly ARs (days) and lower and middle tropospheric specific humidity (g/kg) anomalies (from four days before to five days after the outbreak of ARs) near the North Pole (**a**) and southwestern Alaska (**b**) regions from 1979 to 2019 (all data are detrended and climatological seasonal cycles are removed).

Line 230-233: This commentary is confusing to me, you say that figures 3a and 1b are similar (I agree here) but that this is similar to figure 3b, which I do not see. Figures 3a and 1b show increases of humidity everywhere while humidity related to ARs (figure 3b) shows negative trends so can you explain this a little more. Furthermore, is a sum of Figs 3a and 3b result in fig1b meaning that you get increase trend of humidity everywhere.

Figures 1b, 3a, and 3b represent the total humidity trend, humidity trend unrelated to ARs, and humidity trend related to ARs, respectively. You are right that the total humidity trend (Figure 1b) is equivalent to the sum of Figures 3a and 3b. To enhance clarity, we have rephrased the sentence accordingly (Lines 262-265).

Line 241: I do not understand this, you don't actually look at AR changes in CMIP6 models but just humidity, Z200 and temperature?

This is a typo, it should be Supplementary Fig. 5a in the revision. We corrected it (Line 274).

Line 270: Are you able to actually show the wind anomalies in Figure 4b rather than infer based on the Z200 circulation?

Since the winds in the model are nudged from ERA5, they are exactly the same as those from ERA5. We have included this information for reference. We have shown both wind anomalies and Z200 anomalies between nudging run and control run in Fig. 4b and modified the figure caption in the revision (Fig. 4b, Lines 828-826).

Lines 287-293: Have you plots made to show this? If so, reference them as I am not sure what you are talking about/showing here.

In the original version, we presented the trends of ARs and Z200 in observations (Figs. 1a, 1c), along with the ARs trend of the 40-member CESM2 ensemble mean (Supplementary Fig. 5b). Here, we have included the Z200 trend of the 40-member CESM2 ensemble mean in the supplementary material as Supplementary Fig. 12 (Fig. Rsp4) and reordered the numbering of supplementary figures accordingly. In this revision, we refer to these figures within the main text (Line 327).

Figure Rsp4. Linear trends of summer (JJA) Z200 (m/decade) in the Arctic based on 40-member CESM2 simulations (40-CESM2) from 1979 to 2019. The trends of Z200 are significant across all grid points at the 95% confidence level.

Line 295: What is the size of the subgroup?

There is a total of 40 members. Subgroups, each comprising 8 members, were selected based on AR frequency trends that exceeded the 80th percentile of simulated AR trends for each specific region. We have provided clarification on this issue in the revision (Line 332).

Line 303: You state here that you only use 2 of the 40 members but don't state this in the figure caption so just want to check what you are actually showing in figure 5c and d.

Thank you for your suggestion. We have included additional clarification in the Figure 5c caption to address this point that only two members are shown (Line 841).

Line 309: You have shown previously that the models do not do a good job replicating the trends of AR and Z200 and yet here are saying that they agree with your results and so I am confused about this?

What we meant is that when using the ensemble mean forced solely by anthropogenic forcing, the model does not perform well since it cannot accurately simulate observed wind patterns. However, when we consider the role of internal variability, we can better interpret observations using model results. We have rephrased some of the text to eliminate any potential confusion (lines 345-349).

Figures

Supp fig 5 and 8: In my version of documents, these plots are cut off and so difficult to attest to claims made about these figures in your manuscript.

My apologies for our oversight. We have double-checked to ensure that there are no issues with our plots in the supplementary figures.

Methods:

Line 389: Be specific here, what are the anthropogenic forcings that are held constant?

Thank you for your suggestion. The anthropogenic forcings that are held constant refer to the aerosol concentrations from multiple sources, solar constant, ozone concentration, and all other necessary forcings used in the CMIP6 protocol. The CO₂ values for the years 2000 (Exp: WIN) and 2020 (Exp: WIN+CO₂) are 370 ppm and 415 ppm, respectively. We have clarified this in the text (Lines 447-449).

Line 416: I would just note that you are examining the specific humidity anomalies here, even though it is mentioned in following line, will avoid any confusion about what is meant.

Thank you for your suggestion. To emphasize this, we have added the following sentence in the method part: 'For brevity, in some parts of the text, we use "humidity" and "specific

humidity" interchangeably.' (Lines 407-408).

Much more detail is needed about the nudged model specifically demonstrating what the nudged field looks like.

A new paragraph has been added to describe how we nudge winds in the model. In brief, we apply a constrained nudging of winds during the summer months (JJA) in the Arctic region, from the surface to the TOA. This wind pattern remains constant and mirrors the long-term wind trends observed over the past 41 years. The objective is to understand how ARs respond when this anomalous wind pattern is incorporated into the model. We have shown the nudged 200 hPa wind simulation in the Figure 1c (vectors) and clarified this in the methods section (line 437).

Significance of correlation, you never explain why it is important to account for autocorrelation in the time series.

The assumption for a significance test of correlation, according to the principal of the T-test, is that the time series used to calculate the correlation consist of completely independent samples characterized by white-noise processes (Wilks et al., 2011). However, if the data contain strong auto-correlation (indicating some red-noise process), this assumption may not hold well. To address this issue, it is necessary to consider the effect of effective sample size when testing the significance of correlation using conventional T-tests (Bretherton et al. 1999). We have clarified this point in the revision (Lines 504-510).

References:

- Athanase, M., Schwager, M., Streffing, J., Andrés-Martínez, M., Loza, S. N., & Goessling, H. F. (2022). Impact of the atmospheric circulation on the Arctic snow cover and ice thickness variability. *Geosci. Model Dev.* 6, 849 – 859
- Baxter, I., & Ding, Q. (2022). An optimal atmospheric circulation mode in the Arctic favoring strong summertime sea ice melting and ice–albedo feedback. *Journal of Climate*, 35(20), 6627-6645.
- Bintanja, R., & Selten, F. M. (2014). Future increases in Arctic precipitation linked to local evaporation and sea-ice retreat. *Nature*, 509(7501), 479-482.
- Blackmon, M. L. (1976). A climatological spectral study of the 500 mb geopotential height of the Northern Hemisphere. *Journal of the Atmospheric Sciences*, 33(8), 1607-1623.
- Boisvert, L. N., & Stroeve, J. C. (2015). The Arctic is becoming warmer and wetter as revealed by the Atmospheric Infrared Sounder. *Geophysical Research Letters*, 42(11), 4439-4446.
- Bokuchava, D. D., & Semenov, V. A. (2021). Mechanisms of the early 20th century warming in the Arctic. *Earth-Science Reviews*, 222, 103820.
- Bretherton, C. S., Widmann, M., Dymnikov, V. P., Wallace, J. M., & Bladé, I. (1999). The

- effective number of spatial degrees of freedom of a time-varying field. *Journal of climate*, 12(7), 1990-2009.
- Choi, H. J., Min, S. K., Yeh, S. W., An, S. I., & Kim, B. M. (2023). Seasonally distinct contributions of greenhouse gases and anthropogenic aerosols to historical changes in Arctic moisture budget. *npj Climate and Atmospheric Science*, 6(1), 189.
- Dacre, H. F., Clark, P. A., Martinez-Alvarado, O., Stringer, M. A., & Lavers, D. A. (2015). How do atmospheric rivers form?. *Bulletin of the American Meteorological Society*, 96(8), 1243-1255.
- Ding, Q., Schweiger, A., L'Heureux, M., Battisti, D. S., Po-Chedley, S., Johnson, N. C., ... & Steig, E. J. (2017). Influence of high-latitude atmospheric circulation changes on summertime Arctic sea ice. *Nature Climate Change*, 7(4), 289-295.
- Feng, X., Ding, Q., Wu, L., Jones, C., Wang, H., Bushuk, M., & Topál, D. (2023). Comprehensive representation of tropical–extratropical teleconnections obstructed by tropical Pacific convection biases in CMIP6. *Journal of Climate*, 36(20), 7041-7059.
- Gentile, E. S., Zhao, M., & Hodges, K. (2023). Poleward intensification of midlatitude extreme winds under warmer climate. *npj Climate and Atmospheric Science*, 6(1), 219.
- Gimeno, L., Vázquez, M., Eiras - Barca, J., Sorí, R., Algarra, I., & Nieto, R. (2019). Atmospheric moisture transport and the decline in Arctic Sea ice. *Wiley Interdisciplinary Reviews: Climate Change*, 10(4), e588.
- Johnsen, S. J., Dansgaard, W., & White, J. W. C. (1989). The origin of Arctic precipitation under present and glacial conditions. *Tellus B: Chemical and Physical Meteorology*, 41(4), 452-468.
- Lavers, D. A., Ralph, F. M., Richardson, D. S., & Pappenberger, F. (2020). Improved forecasts of atmospheric rivers through systematic reconnaissance, better modelling, and insights on conversion of rain to flooding. *Communications Earth & Environment*, 1(1), 39.
- Li, Z., Ding, Q., Steele, M., & Schweiger, A. (2022). Recent upper Arctic Ocean warming expedited by summertime atmospheric processes. *Nature communications*, 13(1), 362.
- Liston, G. E., & Sturm, M. (2004). The role of winter sublimation in the Arctic moisture budget. *Hydrology Research*, 35(4-5), 325-334.
- Liu, C., & Barnes, E. A. (2015). Extreme moisture transport into the Arctic linked to Rossby wave breaking. *Journal of Geophysical Research: Atmospheres*, 120(9), 3774-3788.
- Olonscheck, D., Mauritsen, T., & Notz, D. (2019). Arctic sea-ice variability is primarily driven by atmospheric temperature fluctuations. *Nature Geoscience*, 12(6), 430-434.
- Roach, L. A., & Blanchard-Wrigglesworth, E. (2022). Observed winds crucial for september Arctic sea ice loss. *Geophysical Research Letters*, 49(6), e2022GL097884.
- Serreze, M. C., Barrett, A. P., & Stroeve, J. (2012). Recent changes in tropospheric water vapor over the Arctic as assessed from radiosondes and atmospheric reanalyses. *Journal of Geophysical Research: Atmospheres*, 117(D10).

- Tamarin-Brodsky, T., & Kaspi, Y. (2017). Enhanced poleward propagation of storms under climate change. *Nature geoscience*, 10(12), 908-913.
- Topál, D., & Ding, Q. (2023). Atmospheric circulation-constrained model sensitivity recalibrates Arctic climate projections. *Nature Climate Change*, 1-9.
- Topál, D., Ding, Q., Ballinger, T. J., Hanna, E., Fettweis, X., Li, Z., & Pieczka, I. (2022). Discrepancies between observations and climate models of large-scale wind-driven Greenland melt influence sea-level rise projections. *Nature Communications*, 13(1), 6833.
- Vihma, T., Screen, J., Tjernström, M., Newton, B., Zhang, X., Popova, V., ... & Prowse, T. (2016). The atmospheric role in the Arctic water cycle: A review on processes, past and future changes, and their impacts. *Journal of Geophysical Research: Biogeosciences*, 121(3), 586-620.
- Wilks, D. S. (2011). *Statistical methods in the atmospheric sciences* (Vol. 100). Academic press.
- Yang, H., Lohmann, G., Lu, J., Gowan, E. J., Shi, X., Liu, J., & Wang, Q. (2020). Tropical expansion driven by poleward advancing midlatitude meridional temperature gradients. *Journal of Geophysical Research: Atmospheres*, 125(16), e2020JD033158.

Reviewer #2 (Remarks to the Author):

Review of the manuscript NCOMMS-23-52552 entitled “Role of atmospheric rivers in shaping long-term Arctic moisture variability” by Wang et al.

The manuscript shows the influence of the large-scale atmospheric circulation in modulating the atmospheric rivers moistening impact on the Arctic using both the ERA5 reanalysis and idealised simulations with a climate model, in order to explain the trends in the observed atmospheric rivers frequency. The paper is relatively well written. However, I find that many aspects (methods and results) could be better explained, especially the maximum covariance analysis, on which I am not an expert myself. Therefore, I recommend major revisions of the manuscript.

Here below, are my comments listed in the order of the manuscript followed by a few technicalities.

We thank the reviewer for the constructive comments, particularly the suggestion to provide a more comprehensive explanation of the usefulness of MCA and the interpretation of its results. In the revision, we have thoroughly revised our manuscript to address all raised concerns.

Fig. S1a and line 82: The figure represents a mean frequency of atmospheric rivers (AR frequency), not a mean. Please modify the plot and the caption. Moreover, it is not clear how this AR frequency in the Arctic is defined. Is it the mean of the AR frequency north of 60N? Is it the mean of AR frequency at 60N? In this latter case, are only the poleward ARs considered or are all ARs considered? Please explain better in the caption as well as in the Methods section.

Thank you for this valuable suggestion. We have replotted the figures, rewritten the figure captions, and further clarified the definition of Arctic ARs in both the figure caption and the Methods section (Lines 461-462 and Supplementary Fig. 1). The time series represents the area-weighted mean of AR frequency north of 60°N, where area weights are considered. All AR occurrences north of 60°N are taken into account.

Fig. S1b and lines 82-84: To show that the AR frequency and the jet are shifted northward in summer compared to winter, the authors should have represented the difference between winter and summer and not summer only. This behaviour of the jet is well known, but it maybe less known for the ARs. Please consider adding this plot (difference or superimposition of the winter and summer fields in contours).

In response to this suggestion, we have included the mean AR frequency and jet in winter as Supplementary Fig. 1c for comparison with the variables in summer (Lines 95-96 and Supplementary Fig. 1c). Additionally, the difference between the two fields has been added to better illustrate their shift in the supplementary figures.

Fig. S2: In the caption, (blue fan-shaped covered area, → (blue fan-shaped covered area)

We have made the suggested change to the figure caption of Supplementary Fig. 3.

Lines 115-117: the authors mention “moistening trends”, but they do not show any specific humidity trends, for example. It seems that they assimilate this wording to the AR frequency trend. However, it is not fully correct as ARs are not the only source of humidity in the Arctic. The authors also wrote “the Pan-Arctic moistening trend is the most positive and significant (Fig. S1a)”, but they do not show any significance calculation and they should rather refer to Fig. S2b, which shows a trend, shouldn't they? Ideally, the authors should rather represent the specific humidity trend in summer and winter.

Thank you for your suggestion. In Fig. 1b, we now present the summer-specific humidity trend over the Arctic from 1979 to 2019. Additionally, in response to your feedback, we have included the winter trend of specific humidity as Supplementary Fig. 2. Furthermore, we have adjusted the reference figures in the main text accordingly (Line 132).

Line 130: The authors should also add Western Alaska here because it seems to be one of the most prominent pathways of AR into the Arctic in summer, with the largest AR mean frequency north of 60N as seen in Fig. S2a.

Thanks for your suggestion. We have commented on western Alaska trends in the text (Line 149).

Line 133: Why using the geopotential at 200 hPa? It is in the stratosphere at those high latitudes. Could the authors explain why they use this pressure level and not at a lower level, such as 400 or 500 hPa?

The reviewer is correct. In the Arctic during JJA, the tropopause typically resides around 300 hPa, which is relatively high compared to other seasons. Additionally, the circulation trend over the Arctic in summer exhibits a barotropic structure, with its long term geopotential height trends more clear at either 300 hPa or 200 hPa than at lower altitudes. Therefore, we utilized Z200 to represent changes in atmospheric circulation in the upper troposphere. Furthermore, we conducted calculations using Z300 or Z500, revealing similar trends to those observed at 200 hPa.

Line 133: Fig. 1b-1c → Fig. 1b-d (to including panels b, c, and d)

We reorganized these sentences (Line 150-154).

Line 136: The authors should describe the methodology of the maximum covariance analysis in the Methods section and what results it will give so that readers not familiar with this method can understand it.

Thank you for your suggestion. We have included additional explanations regarding MCA in the Methods section (Lines 496-506).

MCA is conceptually similar to principal components analysis (PCA) but is applied to two different data fields. Consider a 2-D (spatial and temporal) data field $X(m,t)$, where m represents the number of total spatial points and t represents the total temporal steps. In PCA, we calculate eigenvectors of the covariance matrix $X \times X'$ (transpose matrix of X), where each element corresponds to the covariance of any pair of two spatial points in X .

In MCA, we work with two different 2-D fields, $X(m,t)$ and $Y(n,t)$, where m and n represent spatial points and t still represents the temporal dimension. MCA aims to compute a singular vector decomposition of $X \times Y'$, where each element in the matrix represents the covariance of a pair of points from X and Y , respectively (one spatial point from X and one spatial point from Y). This enables MCA to identify the dominant coupled modes between variations in X and Y . Due to its widespread use in climate and meteorology since its introduction in the 1990s, we provide a concise explanation in the text without going into more extensive detail.

Lines 130-137 could be written more concisely. Consider rewriting.

We have rewritten it as you suggested. It now reads as follows:

“Patterns of long term variations in AR frequency reveal notable similarities to the observed trends in large-scale JJA circulation (Fig. 1c). Upper-level geopotential heights significantly increase across western Greenland, northern Europe, and eastern Siberia, but over the North Atlantic Ocean, Alaska, and central Siberia the increase is relatively small (three regions are shown in Supplementary Fig. 3e). In line with these atmospheric circulation trends, both specific humidity and temperature display pronounced rises over western Greenland, northern Europe and eastern Siberia (Fig. 1b and 1d)” (Lines 150-166).

Lines 138-144: What about adding SSP5-8.5 to complete the historical simulations and calculate the trend over the exact same period as in the reanalysis (same as what has been performed for CESM2 in Fig. S4b it seems)? Or what about using only 1979-2014 in the reanalysis to match the CMIP6 historical period? A few more or less years can make a difference when calculating trends (see for example Fig. 2 of Outten et al. 2022 <https://wcd.copernicus.org/articles/4/95/2023/wcd-4-95-2023.pdf>).

We have used the three variables only from 1979 to 2014 in the reanalysis to calculate the trends, which are consistent with the results from 1979 to 2019 (Fig. Rsp5 vs Fig. 1a-1d). We also added an analysis of the trend of variable Z200 under the SSP5-8.5 scenario. From 2015 to 2099, the magnitude of the trends became significantly larger, with a similar spatial pattern as that from the historical simulation over the period of 1979-2014 (Fig. Rsp6 vs Supplementary Fig. 4b). Thank you for this suggestion.

Figure Rsp5. a-d, Linear trends of JJA ARs (days/month/decade) (a), lower and middle tropospheric (surface to 500 hPa average) specific humidity (g/kg/decade) (b), 200 hPa geopotential height (Z200) (m/decade) (c), and tropospheric (surface to 200 hPa average) air temperature (K/decade) (d) in the Arctic from 1979 to 2019. Black dots in a-d denotes statistically significant trends at the 95% confidence level.

Figure Rsp6. Linear trend of summer 200 hPa geopotential height (Z200) (m/decade) (a) in the Arctic from 2015 to 2099 based on 40 CMIP6 models under SSP 5-8.5 scenario. The trend of Z200 significant everywhere at the 95% confidence level.

Line 142-143: why not using the same vertical extent for all variables? Moreover, these vertical extents are different than the one used to calculate the IVT (line 399). Can the authors justify their choice?

Thank you for your suggestion. In fact, our results are not sensitive to how we conduct vertical integration, given that these features typically exhibit quite a barotropic structure (uniformly vertically). However, we provide specific reasons for why, for some plots, we only focus on certain levels or average for certain levels.

For the humidity and air temperature variables, we focus on different vertical extents: an average from the surface to 500 hPa, and an average from the surface to 200 hPa, respectively. We find changes in Z200 are quite similar to changes in geopotential heights at other levels. For the simplicity of presentation, we use Z200 to analyze trends in upper-level atmospheric circulation. Moreover, since most of the moisture in the atmosphere is concentrated in the middle and lower troposphere, we integrate humidity from the surface to 500 hPa to represent variations in atmospheric moisture (Sun and Lindzen, 1993). Moisture content above 500 hPa is almost zero in the Arctic. Additionally, as Z200 change also represents a thickness change (equivalent to temperature) of the entire air column from 200 hPa to the surface, we integrate vertically from the surface to 200 hPa when analyzing air temperature changes. When calculating IVT, we integrate moisture across the entire troposphere as required by the existing AR scheme. In fact, using IVT integrated from 500 hPa to the surface would not change our result.

Line 144: Could the authors add a measure of significance on Fig. S3? An easy one could be the number of models agreeing on the sign of the trend. Moreover, are there some CMIP6 models in better agreement with ERA5 than others? Could the answer to this question be mentioned in the manuscript?

We have conducted a significance test on the trend of the variables in the Supplementary Fig. 4 (Supplementary Fig. 4 in the revision). Since all grid points across all figures are significant, we made note of this in the figure caption.

We also examine each individual model among the 10 models used, revealing that most simulations fail to capture the observed trend of JJA Z200 in ERA5. The results across the 10 models exhibit significant diversity (Fig. Rsp7).

Figure Rsp7. Linear trend of summer 10 models (a-j) 200 hPa geopotential height (Z200) (m/decade) in the Arctic from 1979 to 2014.

Fig. S4a: Why is this figure the mean of only 10 CMIP6 models when Fig. S3 includes 40 models? Could Fig. S3 be redrawn with the 10 same models?

We used monthly data from 40 CMIP6 models to generate Supplementary Fig. 4 (Supplementary Fig. 4 in revision). However, when calculating ARs, daily or even higher time-resolution specific humidity and wind fields are necessary. Here, only 10 models provide such high temporal resolution data. Therefore, when examining changes in ARs, we only considered data from these 10 models. In response, we have added a sentence to the Methods section to clarify this (Lines 415-418).

In the caption: “and 40-member” → and (b) 40-member

We have made the suggested change in the caption of Supplementary Fig. 5.

Line 148: What are the SST changes the authors mention here? Please be more precise. Why would it be the only factor influencing the AR trend found in CMIP6?

Since this is not a key point of our paper and to avoid further confusion, we removed this part of the sentence.

Line 165: What are “the long term changes of ARs and circulation”? Do the authors mean the trend?

Here we refer to the leading mode of MCA for the interannual variation of two variables, which is similar to the spatial patterns of those long-term trends. We modified the sentence to 'the long-term trends of ARs and circulation' (Lines 189-190)

Line 181: Shouldn't it be MCA1 ARs-Z200 instead of MCA1-ARs or MCA1-Z200 since the maximum covariance analysis is performed using both the AR frequency and Z200? Do the authors expect different results by using MCA1-ARs instead of MCA1-Z200? Please clarify the MCA methodology.

The leading mode of MCA (which contains two spatial patterns for each field) has two time series (referred to as MCA1-ARs and MCA-Z200, respectively). As these two time series are highly correlated (Corr. = 0.83), utilizing either of the time series as an index to calculate correlations with other variables yields very consistent results. In response, we added a section in the Methods to introduce the MCA (Lines 496-506).

Line 195: Greenland → the Western Greenland region

We have made the suggested change (Line 224).

Lines 196-198: Could the authors add a reference to this statement?

We have added a reference: Van den Broeke et al. 2016 (Line 227).

Line 199: What do the authors mean with “daily AR frequency”? Aren't the daily fields binary masks (0-1 fields) of atmospheric rivers?

Yes, this number indicates the percentage of the one-day period (24h) during which the grid box is governed by ARs.

Why do the authors use both CESM1 and CESM2 in their study? Would it be possible to use only one model version so that the results of their two analysis are comparable? Do the

authors expect the two versions of CESM to give similar results for their idealised experiments? This uncertainty could be acknowledged somewhere in the manuscript.

Since the data (6-hour 3-D U, V, and specific humidity) used to calculate ARs is not available in CESM1-LEN (which was conducted almost 8 years ago), but is available in CESM2-LEN, we strategically utilized both CESM1 and CESM2 in this study. For our fingerprint analysis, we relied on CESM2-LEN since ARs can be obtained from this dataset. However, for our nudging run, given the more efficient calculation speed of CESM1 compared to CESM2, we relied on CESM1. Actually, CESM2 did not exhibit significant changes compared to CESM1 when the nudging scheme was employed on winds (Topál et al., 2023). Thus, we believe that our results were not sensitive to the choice of model. We have clarified this point in the paper. (Lines 445-447)

Lines 206-207: Isn't that result coming directly from the MCA? Would the authors expect another result? Also, the figure referred to should be Fig. 2d not 2c, right?

The results are derived from MCA after detrending the data and removing climatological seasonal cycles. To facilitate easier comparison of the time series, we standardized them after applying MCA. We were not surprised to see the MCA result at the beginning, as the coherent spatial features of trends in these variables already suggested that these fields are closely coupled on some time scales. Thanks for bringing the figure reference error to our attention. It has been corrected. (Line 236)

Figure 2d: - Could the authors give more details on what is represented on panel d? Why is the y-axis label saying "Standard deviation" whereas the caption text says "standardized time series"? Please clarify.

- Moreover, the straight horizontal parts in the blue curve highlight the days with no AR. Is that right? Isn't it an issue to perform the MCA with some time series having 0s?

- Finally, it is not clear to me if this time series (panel d) is for the central point of the region or an average over the region or something else. Please add this precision.

Figure 2d depicts the standardized time series of the leading MCA mode of ARs and lower and middle tropospheric specific humidity over western Greenland in summer 2012.

The time series in Figure 2d illustrate the temporal changes in the spatial pattern of the leading MCA mode between ARs and specific humidity over western Greenland in summer 2012.

In the revision, we have included a new section in the Methods to introduce the MCA method (Lines 496-506).

Lines 207-208: Again, what does the value of the time series peak on 11 July (Fig. 2d) mean? I understand that the fact above 0 values mean that there is an atmospheric river and that many days around the 11th experienced an atmospheric river. Is that correct?

You are correct. When there is ARs activity at some grid points over western Greenland, the ARs value will be higher than 0. However, on July 11, the ARs activity area reached its maximum over western Greenland, and the humidity anomaly also reached the highest. We have marked July 11 in red font and added a sentence in the figure caption of Fig. 2 to clarify this point (Lines 815-816).

Line 209: remove).

We have made the suggested change. (Line 238)

Figure 2c and Figure S7: Why do the two figures have different y-axis label (AR change (days) for Fig. 2c and AR anomaly (case) for Fig. S7). Please adjust and make it consistent (both the label and the unit).

Thanks for your suggestion. The label should be “days” in Supplementary Fig. 10 (Supplementary Fig. 10 in the revision). We have made the correction.

Lines 228-230: The methodology is not clear at all to me. I suggest a major rewriting of the method section for this analysis.

We have rephrased many parts of the methodology section to improve its readability (Lines 482-495).

Lines 233-234: How are those trends calculated? Please explain in the text.

Following our 'spatiotemporal extension' method, we extract 6-hourly specific humidity data related to and unrelated to AR activity during summer over the Arctic. We then aggregate these datasets into JJA averages and calculate their respective trends from 1979 to 2019. In the revised version, we have provided a more detailed explanation of this approach (Lines 482-495).

Figure 3: My understanding is that Fig. 3b is obtained by subtracting Fig. 3a from Fig. 1b. Is that correct? In Eastern Siberia, it looks like the sum of Fig. 3a and b would be higher than Fig. 1b. Could the authors check? In any case, please add some explanations to the methods (as also written above).

Fig. 1b, Fig. 3a, and Fig. 3b depict the trends of total humidity, humidity unrelated to ARs, and humidity related to ARs, respectively. Therefore, the sum of Fig. 3a and Fig. 3b equals Fig. 1b. We have verified these results to be accurate. Additionally, we have redrawn Fig. 1b, and below, we present the new Fig. 1b along with the sum of Fig. 3a and Fig. 3b as Fig. Rsp8.

Figure Rsp8. Linear trends of JJA lower and middle tropospheric (surface to 500 hPa average) specific humidity (g/kg/decade) associated with ARs (left panel), and the sum of specific humidity unrelated and related to ARs (right panel) in the Arctic from 1979 to 2019. Stippling indicates statistical significance?

Lines 241-242: I am sorry, but I do not understand how the authors can say that from Fig. S3a. Could the authors elaborate a bit here?

We apologize for our figure referencing error, which should be Supplementary Fig. 5a. We have corrected this error in the revision. (Line 274)

Line 254: Does the addition of the anomalies introduce noise or discontinuities in the wind field?

Since this method is adapted from a widely used assimilation method developed by NCAR DART team (<https://journals.ametsoc.org/view/journals/clim/25/18/jcli-d-11-00395.1.xml>), we do not anticipate significant errors in our calculations using this approach. However, it's still possible that some subtle noise may be introduced at certain levels, but it should be very trivial given the focus on the long-term trends in our results.

Line 276: How small is this wind change?

Wind changes in the model between the nudging run and the control run are approximately 0.5-1 m/s (see Fig. 4b, vectors), which is relatively small compared to the climatology wind speed of JJA, ranging from 6-12 m/s (see Supplementary Fig. 3c, vectors) within the Arctic.

Line 291: "...and again, and reinforcing..." needs rewriting.

We have rewritten the phrase as recommended (Line 328).

Line 366: add (Z200) after "Arctic atmospheric circulation"

We have made the suggested change (Line 407).

Lines 367-368: Why do the authors use both 6-hourly and monthly reanalysis data? Moreover, since data from models are available daily, why not also use daily reanalysis, instead of 6-hourly? It looks inconsistent, especially as the ARs are detected from daily fields for the models but 6-hourly fields for the reanalysis.

Thank you for your comment. We primarily utilize monthly reanalysis data to calculate the trend of variables in our study (or for MCA). However, when identifying AR variability, 6-hour data is ideal due to the transient and extreme nature of ARs, where higher temporal resolution data can better capture their characteristics. Unfortunately, only a few CMIP6 models provide daily data (3-D U, V, and specific humidity over multiple representative vertical levels) for the public, and 6-hour data for these variables are not available. Consequently, we have to employ a remedial approach using daily data to detect ARs. In response, we have added sentences to the Methods section to clarify our approach (Lines 412-418).

Line 398 (and others, e.g. 425, 426): “grid” → grid point

Following the Oxford dictionary, the noun grid means “An arrangement of parallel bars with openings between them”. Therefore, the authors seem to misuse this noun. Please correct where appropriate.

We appreciate the reviewer’s recommendation and have made several corrections (Lines 136, 454, 484).

Line 410: “lag” or last?

Here, using ‘last’ is more appropriate, and we have made the suggested change (Line 473).

Lines 422-423: “Initially ... algorithm”. This sentence is not understandable to me. Please adjust and clarify.

Thanks for your suggestion. We have made the following modification:

‘Initially, grid points that have AR activity are determined by the 'shape' of ARs as defined by the detection algorithm.’ (Lines 484-485).

Line 424: “extended spatiotemporally”. The spatial extension is of 6 degrees, but how long in time is the AR impact extended?

Water vapor anomalies associated with ARs are observed to precede the onset of ARs by approximately 1 day and to lag behind the termination of ARs by approximately 1 day (see Figs. 2c, Supplementary Fig.10, and Supplementary Fig.11). To account for these lead-lags, we have extended the time window by one day on both ends. Additionally, we have included

some text to elucidate this process (Lines 487-490).

Line 426: What do the authors mean with “time point”?

This should be ‘time,’ and we have made the suggested correction (Line 492).

Lines 428-429: How can the AR impacts be removed? Please add an explanation.

Following the 'spatiotemporal extension' method (details see Methods section, lines 480-493), we have differentiated the humidity trends related to ARs from those unrelated to ARs. To make it clearer, we have modified the sentence (Lines 493-495).

To provide a quick overview of our approach, we briefly summarize it here. AR events reaching the Arctic exhibit a sporadic pattern during the JJA season. At each time step (every 6 hours) over grids within the Arctic region, if certain grids are identified as influenced by ARs based on our AR detection scheme, we mask these grid points in subsequent calculations. This screening process is applied across all 40 years at 6-hourly intervals. The processed 6-hour specific humidity data are aggregated for each JJA season. We then recalculate the linear trend of this newly constructed JJA-specific humidity. Through this process, we can isolate the linear trend of specific humidity without the influence of ARs.

Figures:

- Please represent the coastlines in black. They are barely visible in gray.

Thanks for your suggestion. To distinguish it from the black dots of the significance test, we used darker gray for the map outline.

- Figure 1 would benefit a lot from larger panels with for example only 3 panels on the top row.

We appreciate the suggestion and have modified the panels accordingly (Fig. 1).

- How is the significance calculated (the black dots in the figures)?

The significance of trend was calculated by Mann-Kendall Test (Mann 1945; Kendall 1948; Gilbert 1987; and Hamed and Rao 1998). The Mann-Kendall Test analyzes difference in signs between earlier and later data within a time series. The idea is that if a trend is present, the sign values will tend to increase constantly, or decrease constantly. The detail of the calculation can be found here:

(https://vsp.pnnl.gov/help/vsample/design_trend_mann_kendall.htm)

Technicalities (probably not exhaustive):

Line 53: altered Arctic → altered the Arctic

We have made the suggested correction (Line 53).

Line 78: move eastward → move (north)eastward

We have made the suggested correction (Line 84).

Line 135: in the Fig. S2b, Fig. 1c → in Fig. S2b, Fig. 1c

We have made the suggested correction, which should be Supplementary Fig. 3e in the revision (Line 154).

Line 399: $0.25^\circ \times .25^\circ \rightarrow 0.25^\circ \times 0.25^\circ$

We have made the suggested correction (Line 456).

Line 442: is → are

We have made the suggested correction (Line 525).

Line 655: The dotted → Black dots

We have made the suggested correction (Line 792).

Lines 681, 690, 699: The dotted grids → Black dots

We have made the suggested corrections (Lines 821, 835, 844. Supplementary Fig. 5, Supplementary Fig. 6, and Supplementary Fig. 13).

References:

Sun, D. Z., & Lindzen, R. S. (1993). Distribution of tropical tropospheric water vapor. *Journal of Atmospheric Sciences*, 50(12), 1643-1660.

Wallace, J. M., Smith, C., & Bretherton, C. S. Singular value decomposition of wintertime sea surface temperature and 500-mb height anomalies. *J. Clim.* 5, 561-576 (1992).

Mo, R. (2003). Efficient algorithms for maximum covariance analysis of datasets with many variables and fewer realizations: A revisit. *Journal of Atmospheric and Oceanic Technology*, 20(12), 1804-1809.

Mann, H. B. (1945). Nonparametric tests against trend. *Econometrica: Journal of the econometric society*, 245-259.

Kendall, M. G. (1948). Rank correlation methods.

Gilbert, R. O. (1987). Statistical methods for environmental pollution monitoring. John Wiley & Sons.

Hamed, K. H., & Rao, A. R. (1998). A modified Mann-Kendall trend test for autocorrelated data. *Journal of hydrology*, 204(1-4), 182-196.

REVIEWERS' COMMENTS

Reviewer #1 (Remarks to the Author):

This manuscript explores the role of large-scale circulation variability on AR frequency within the Arctic in observations and large-ensemble and bespoke models forced with wind anomalies. The authors find that the increased trend of AR and moisture is related to the circulation over the Arctic bringing more AR and moisture into Greenland, Siberia and northern Europe.

I want to thank the authors for the time and energy spent to comprehensively address my comments and questions and for all the additional work that went into the manuscript. I have only minor comments on this iteration of the manuscript.

Line 53: Can you give examples of how moisture increase has altered the Arctic hydrology and cryosphere?

Line 170: Rogue underscore

Line 171: Have -> Has

Line 321: Is the ensemble mean of the “two fields” you talk about here supplemental figure 12? I would just state that first and then mention the other figures that you are comparing it too.

Lines 379-381: Seems like a word or phrase is missing or perhaps needs reworded to emphasize that the increase in moisture can impact sea ice if that is the purpose of these lines.

Line 421: Outline what WIN and WIN+CO2 are before you reference them mid-sentence. The text in lines 274-294 that describe the models in detail I feel should be in the methods section

Line 427: numerically incorporating? Should it be numerically incorporate?

Line 456: Would it be better to say that the IVT anomalies lag behind rather than “last”?

Line 491: Consist -> Consists

Figures 1 and supplemental figure 4 – I think it might be good to have these on the same colorbar, that way it will really emphasise the lack of change in CMIP compared to ERA5

Reviewer #2 (Remarks to the Author):

Review of revised manuscript NCOMMS-23-52552A entitled "Role of atmospheric rivers in shaping long term Arctic moisture variability" by Wang et al.

The authors have addressed most of my previous comments. Overall, I think that the manuscript could still be improved, especially the first part that is slightly difficult to read. The authors can find my comments on their revised version here below, with my specific comments and minor/technical comments. In light of these comments, I suggest to revise again the manuscript.

Specific comments:

Line 39: What do the authors mean with "wetting trends"? More moisture or more precipitation? Please clarify in the manuscript.

Line 42: "AR activity, driven by extreme synoptic weather systems" Not all ARs are associated with very strong cyclones or are they? Can't they be associated with more moderately intense cyclones? Please adjust text.

Line 70-74: This is a confusing sentence. It seems that the authors just want to say that the anticyclonic anomaly favours adiabatic warming, hence melting of ice. Please rephrase.

Lines 77-81: "While the CC relationship governs the temperature-humidity relationship on a global scale, internal atmospheric circulation variability at various time scales also plays a key role in the Arctic, regulating the distribution and transport of moisture across a wide range of time scales through its effects on weather systems including intense storms such as Arctic cyclones and atmospheric rivers (ARs)." This is a very long and not clear sentence. I suggest to split it in two and to clarify the message. Note also the repetition of "relationship" at the beginning of the sentence.

Line 80: "intense storms such as Arctic cyclones" Are Arctic cyclones intense storms? Are they stronger (in pressure or in wind intensity) than mid-latitudes cyclones?

Lines 83-85: "ARs typically form as a result of the interaction between a cold front, which transports a substantial amount of water vapor in the warm sector, and a warm front." I do not think this is correct. ARs are collocated with the cold front, which separates the warm sector from the cold sector. It is not linked to the warm front.

Line 92: "more AR propagation into the Arctic" > more ARs propagating into the Arctic

Lines 134-137: this part belongs to the Methods.

Line 177: "AR characteristics" The authors only analyse the frequency of atmospheric rivers. Therefore, I

suggest to change it to “AR frequency”.

Lines 197-199: I have a bit of trouble with the “steering forcing” mentioned here. By definition, the atmospheric rivers, via the integrated water vapour transport, are highly dependent on the wind. Therefore, it is normal that the atmospheric rivers are associated with the wind. In my view, the atmospheric rivers which are most often associated with cyclones at those latitudes rather depend on what drives the cyclones in one direction or another. Please modify this sentence.

Fig. 2d: Please explain somewhere (main text or figure caption) what the horizontal blue lines in this time series mean.

Lines 254-255: 1) The trend does not change with time in this study. I think the authors mean here long-term trends in Arctic ARs frequency.

2) Please acknowledge that there could be other sources of moisture in the Arctic not linked to atmospheric rivers, so that only most of the trend can be explained by changes in moisture.

Line 256: What are “these impacts”? Please clarify here.

Line 259: I would not say that the AR activity is “rather uniform”. There are maxima west of Greenland, Siberia and in the North Atlantic.

Line 269: Why don't the authors also comment CESM2 here (Fig. S5b)?

Lines 305-306: “a small wind field change”: is it smaller than the northerly wind in line 301?

Line 353: “ARs are primarily driven by the atmosphere's baroclinic instability of the mean state” I do not think this is true. It is the cyclone to which the AR “belongs” that is controlled by baroclinic instability. Please change and/or clarify.

Line 354: Could the authors explain here why atmospheric rivers “exhibit stochastic behaviour on shorter timescales”? “Shorter” than what? Why “stochastic”?

Line 356: “Arctic ARs have exhibited a well-organised increase” I suppose the authors mean an increase in frequency (and not an increase in intensity for example). Please add “increase in frequency”.

Methods: Although it is the main topic of the study, the authors still do not explain how they calculate the trends. Do they use least-square regression, a Theil-Sen estimator, another method? Please add a description of the method in this section.

Figures 1d and 2d: Again, why is the y-axis label “Standard deviation”? Is it a spatial standard deviation? To my understanding, if the authors were using EOFs, they would represent here the principal components (which are not a standard deviation). Do the authors mean expansion coefficients? Please change the y-axis label with the appropriate name or explain better what is represented with this time series.

Technical and minor comments:

Line 31: "This" → The

Line 60: "Arcitc" → Arctic

Line 81: "characterized as" → characterized by

Line 95: "leading dramatic impacts on the local climate" → impacting the local climate

Supplementary Figure 1b,c: "Climatological mean ARs" → AR frequency, also in the legend "AR mean" does not mean anything. Please change to AR frequency.

Line 105: "consequently enhancing the more occurrence of ARs there" → consequently increasing the occurrence of ARs there

Line 109: "and low-frequency" → and the low-frequency

Line 116: 68. -> 68.

Line 129: "the role of ARs in contributing to recent" -> the contribution of ARs to the

Line 134: "To study year-to-year" -> To study the year-to-year

Lines 137, 138: "grid" -> grid point

Line 171: "have" -> has

Line 173: "model" -> models

Line 184: "its spatial patterns clearly exhibit" -> its spatial pattern clearly exhibit

Line 190: "may physically connected" -> may be physically connected

Line 194: "in climatology" -> in the climatology

Fig. 2c: The x-axis label is missing. Should be "Days" as in the Supplement.

Line 251: "The similar" -> A similar

Line 262: Please add that Arctic is north of 60°N here.

Line 272: "that large-scale circulation" -> that the large-scale circulation

Line 278: "to top of" -> to the top of

Line 297-298: Remove ", which is characterized by a zonal wave number 2 structure,"

Line 300: Where is "the high pressure"? Over Canada/Canadian archipelago?

Lines 302-303: Is the response of northern Europe really above 0.5 days per month?

Lines 336, 346, 347, 365: "AR": AR frequency

Line 400: "6-hour" -> 6-hourly

Line 470: "spatiotemporally" -> spatially

Lines 474, 475: "grids" -> grid points

Caption of Figure 1:

Lines 783, 788, 789: "ARs" -> ARs frequency

"lower and middle" -> lower to middle

Lines 785-786: remove ", or m/s/10a"

Line 789: "denotes" -> denote

Line 790: "for wind" -> for the wind

Caption of Figures 4 and 5:

Line 829, 839, 840: "ARs" -> AR frequency

Please also check the captions of the Supplement Figures.

REVIEWERS' COMMENTS

Reviewer #1 (Remarks to the Author):

This manuscript explores the role of large-scale circulation variability on AR frequency within the Arctic in observations and large-ensemble and bespoke models forced with wind anomalies. The authors find that the increased trend of AR and moisture is related to the circulation over the Arctic bringing more AR and moisture into Greenland, Siberia and northern Europe.

I want to thank the authors for the time and energy spent to comprehensively address my comments and questions and for all the additional work that went into the manuscript. I have only minor comments on this iteration of the manuscript.

We appreciate the reviewer for their valuable suggestions, which provide us with a new opportunity to improve our manuscript. We hope that the modifications we have made in response to the comments and suggestions will meet the reviewer's expectations.

Line 53: Can you give examples of how moisture increase has altered the Arctic hydrology and cryosphere?

In the Arctic, the increase in moisture in the atmosphere regulates hydrology and the cryosphere in several ways, including changing atmospheric radiative properties by altering moisture content and cloud characteristics, surface albedo conditions by depositing different types of snow on the surface, and the mass balance of ice sheets and glaciers through precipitation processes, among many others (Bring et al., 2016; Vihma et al., 2016). We have added relevant references covering these aspects to support that sentence (Line 52).

Line 170: Rogue underscore

We have made the suggested correction (Line 165).

Line 171: Have -> Has

We have made the suggested correction (Line 166).

Line 321: Is the ensemble mean of the “two fields” you talk about here supplemental figure 12? I would just state that first and then mention the other figures that you are comparing it too.

Regarding that part, we compared the trends of ARs and Z200 between observations and the ensemble mean. To improve clarity of that sentence, we have replaced 'two fields' with 'ARs and Z200 trends' and reorganized how we reference these figures in supporting that statement (Lines 318-319).

Lines 379-381: Seems like a word or phrase is missing or perhaps needs reworded to emphasis that the increase in moisture can impact sea ice if that is the purpose of these

lines.

We have rephrased the latter part of this sentence as follows: ‘... sea ice and land ice.’ (Line 379).

Line 421: Outline what WIN and WIN+CO2 are before you reference them mid-sentence. The text in lines 274-294 that describe the models in detail I feel should be in the methods section.

We have implemented the suggested correction at lines 419-420. As we consider the text from original lines 274-294 crucial for aiding readers' understanding of the subsequent section, we have retained the majority of it.

Line 427: numerically incorporating? Should it be numerically incorporate?

It should be ‘incorporate’. We have made the suggested correction (Line 426).

Line 456: Would it be better to say that the IVT anomalies lag behind rather than ‘last’?

Here, we want to emphasize that the moisture anomalies will persist for one day after ARs end. Therefore, using 'last' may be more appropriate in that context.

Line 491: Consist -> Consists

It should be ‘consist’ to agree with the plural subject ‘time series’ of that sentence.

Figures 1 and supplemental figure 4 – I think it might be good to have these on the same colorbar, that way it will really emphasise the lack of change in CMIP compared to ERA5

Thank you for your suggestion. We have redrawn Supplementary Figure 4 using the same color bar as Figure 1 (Supplementary Fig. 4).

References:

Bring, A., Fedorova, I., Dibike, Y., Hinzman, L., Mård, J., Mernild, S. H., ... & Woo, M. K. (2016). Arctic terrestrial hydrology: A synthesis of processes, regional effects, and research challenges. *Journal of Geophysical Research: Biogeosciences*, 121(3), 621-649.

Vihma, T., Screen, J., Tjernström, M., Newton, B., Zhang, X., Popova, V., ... & Prowse, T. (2016). The atmospheric role in the Arctic water cycle: A review on processes, past and future changes, and their impacts. *Journal of Geophysical Research: Biogeosciences*, 121(3), 586-620.

Reviewer #2 (Remarks to the Author):

Review of revised manuscript NCOMMS-23-52552A entitled “Role of atmospheric rivers in shaping long term Arctic moisture variability” by Wang et al.

The authors have addressed most of my previous comments. Overall, I think that the manuscript could still be improved, especially the first part that is slightly difficult to read. The authors can find my comments on their revised version here below, with my specific comments and minor/technical comments. In light of these comments, I suggest to revise again the manuscript.

We deeply appreciate the comprehensive review conducted by the reviewer, recognizing the significant time and effort involved. Therefore, we have carefully integrated all the comments and suggestions into the manuscript, aiming to ensure the paper meets expectations in its current form. We hope the reviewer finds the revisions satisfactory.

Specific comments:

Line 39: What do the authors mean with “wetting trends”? More moisture or more precipitation? Please clarify in the manuscript.

We thank the reviewer for this suggestion. It refers to the upward trend of atmospheric moisture. We have replaced ‘wetting’ with ‘humidity’ (Line 39).

Line 42: “AR activity, driven by extreme synoptic weather systems” Not all ARs are associated with very strong cyclones or are they? Can’t they be associated with more moderately intense cyclones? Please adjust text.

Thank you for your suggestion. Given that ARs are associated with two key features: strong cyclonic rotation and large-scale moisture transport, their occurrence is often driven by strong synoptic-scale systems associated with moisture transport originating from the tropics. This is because the abundance of moisture and the dynamic intensity of low-pressure systems (such as extratropical storms and cyclones) mostly tend to vary in-phase. To refine the sentence, we replaced 'extreme' with 'strong' and added 'mostly' there (Line 41).

Line 70-74: This is a confusing sentence. It seems that the authors just want to say that the anticyclonic anomaly favours adiabatic warming, hence melting of ice. Please rephrase.

We provide additional information before that sentence to improve the readability of both sentences. The new sentence reads ‘...manifested as a long term trend toward the local barotropic high pressure anomaly situated over Greenland over the past four decades.’ (Lines 68-69).

Lines 77-81: “While the CC relationship governs the temperature-humidity relationship on a global scale, internal atmospheric circulation variability at various

time scales also plays a key role in the Arctic, regulating the distribution and transport of moisture across a wide range of time scales through its effects on weather systems including intense storms such as Arctic cyclones and atmospheric rivers (ARs).” This is a very long and not clear sentence. I suggest to split it in two and to clarify the message. Note also the repetition of “relationship” at the beginning of the sentence.

Thank you for your suggestion. We have split the sentence into two parts (Lines 75-79). Additionally, we replaced 'relationship' with 'equation' in the first sentence (Lines 75-77).

Line 80: “intense storms such as Arctic cyclones” Are Arctic cyclones intense storms? Are they stronger (in pressure or in wind intensity) than mid-latitudes cyclones?

Not all cyclones can reach intense storm status. To better reflect this point, we have modified the sentence to read ‘...including extratropical storms such as Arctic cyclones and atmospheric rivers (ARs)’ (Lines 78-79).

Lines 83-85: “ARs typically form as a result of the interaction between a cold front, which transports a substantial amount of water vapor in the warm sector, and a warm front.” I do not think this is correct. ARs are collocated with the cold front, which separates the warm sector from the cold sector. It is not linked to the warm front.

We have removed ‘the warming sector’ from the sentence (Line 83). Thanks for raising this issue.

Line 92: “more AR propagation into the Arctic” > more ARs propagating into the Arctic

We have made the suggested correction (Line 90).

Lines 134-137: this part belongs to the Methods.

We agree. We have relocated this part to Methods section under ‘AR detection’ (Lines 449-452).

Line 177: “AR characteristics” The authors only analyse the frequency of atmospheric rivers. Therefore, I suggest to change it to “AR frequency”.

We have made the suggested correction (Line 172).

Lines 197-199: I have a bit of trouble with the “steering forcing” mentioned here. By definition, the atmospheric rivers, via the integrated water vapour transport, are highly dependent on the wind. Therefore, it is normal that the atmospheric rivers are associated with the wind. In my view, the atmospheric rivers which are most often associated with cyclones at those latitudes rather depend on what drives the cyclones in one direction or another. Please modify this sentence.

We have modified ‘steering forcing’ to ‘governing forcing’ since we aim to emphasize that the JJA mean anomalies provide a relatively long-term background flow in which synoptic-scale ARs can receive some impacts (Line 193). This scenario is very similar to how the summer mean flow regulates the activity of hurricanes over the tropical

Atlantic or western Pacific.

Fig. 2d: Please explain somewhere (main text or figure caption) what the horizontal blue lines in this time series mean.

The horizontal blue line in Figure 2d indicates the absence of AR activity during the period. We added the relevant words in the figure caption to explain this point (Lines 818-819).

Lines 254-255: 1) The trend does not change with time in this study. I think the authors mean here long-term trends in Arctic ARs frequency.

Yes. We have rephrased that sentence as ‘Possibly, changes in long term trends of atmospheric moisture are also mirrored by long term changes in Arctic AR frequency to some extent.’ (Lines 248-250).

2) Please acknowledge that there could be other sources of moisture in the Arctic not linked to atmospheric rivers, so that only most of the trend can be explained by changes in moisture.

Thanks for your comment. We have changed ‘primarily’ to ‘partly’ and ‘to some extent’ to make this point clearer (Lines 248-250).

Line 256: What are “these impacts”? Please clarify here.

This refers to impacts of ARs on moisture changes in the Arctic. We have added a few words there to clarify this point (Lines 251).

Line 259: I would not say that the AR activity is “rather uniform”. There are maxima west of Greenland, Siberia and in the North Atlantic.

Agree. We have modified the sentence as ‘... has increased in most Arctic areas, especially in western Greenland, Siberia, and the North Atlantic.’ (Lines 254-255)

Line 269: Why don’t the authors also comment CESM2 here (Fig. S5b)?

Thanks for raising this comment. We have rephrased the sentence to include CESM2 there (Lines 265-267).

Lines 305-306: “a small wind field change”: is it smaller than the northerly wind in line 301?

We understand your concern regarding our vague use of 'small' in the sentence. We have removed 'small' and added the magnitude (1 m/s) of the wind change in that sentence (Lines 302-303).

Line 353: “ARs are primarily driven by the atmosphere’s baroclinic instability of the mean state” I do not think this is true. It is the cyclone to which the AR “belongs” that is controlled by baroclinic instability. Please change and/or clarify.

Agree. ARs are also influenced by the availability of atmospheric moisture. We changed 'primarily' to 'mainly' since it is still reasonable to use 'mainly' here, as these

frontal low-pressure systems (due to baroclinic instability) serve as the backbone of ARs (Line 350).

Line 354: Could the authors explain here why atmospheric rivers “exhibit stochastic behaviour on shorter timescales”? “Shorter” than what? Why “stochastic”?

AR activity over the synoptic scale appears to be stochastic, given the nature of baroclinic instability and the supply of moisture from the tropics. To avoid any possible confusion generated by this word, we have modified 'stochastic' to ‘highly variable’ (Line 351).

Line 356: “Arctic ARs have exhibited a well-organised increase” I suppose the authors mean an increase in frequency (and not an increase in intensity for example). Please add “increase in frequency”.

We have made the suggested correction (Line 353).

Methods: Although it is the main topic of the study, the authors still do not explain how they calculate the trends. Do they use least-square regression, a Theil-Sen estimator, another method? Please add a description of the method in this section.

We used the least-square regression method to estimate linear trends in this study. We have emphasized this method in the Methods section (Lines 393).

Figures 1d and 2d: Again, why is the y-axis label “Standard deviation”? Is it a spatial standard deviation? To my understanding, if the authors were using EOFs, they would represent here the principal components (which are not a standard deviation). Do the authors mean expansion coefficients? Please change the y-axis label with the appropriate name or explain better what is represented with this time series.

Thanks very much for your suggestion. We have replaced “Standard deviation” with “Standardized time coefficient” in all relevant figures, including Figures 1 and 2, and Supplementary Figures 7, 8, and 9.

Technical and minor comments:

Line 31: “This” → The

We have rewritten the sentence (Lines 31-33).

Line 60: “Arcitc” → Arctic

We have made the suggested correction (Line 58).

Line 81: “characterized as” → characterized by

We have made the suggested correction (Line 79).

Line 95: “leading dramatic impacts on the local climate” → impacting the local climate

We have made the suggested correction (Line 93).

Supplementary Figure 1b,c: “Climatological mean ARs” → AR frequency, also in the legend “AR mean” does not mean anything. Please change to AR frequency.

We have made the suggested correction (Supplementary Figure 1).

Line 105: “consequently enhancing the more occurrence of ARs there” → consequently increasing the occurrence of ARs there

We have made the suggested correction (Line 103).

Line 109: “and low-frequency” → and the low-frequency

We have made the suggested correction (Line 107).

Line 116: 68. -> 68.

We have made the suggested correction, and now it is Ref 69 in the revision (Line 114).

Line 129: “the role of ARs in contributing to recent” -> the contribution of ARs to the

We have made the suggested correction (Line 127).

Line 134: “To study year-to-year” -> To study the year-to-year

We have made the suggested correction (Line 449).

Lines 137, 138: “grid” -> grid point

We have made the suggested correction (Lines 452, 133).

Line 171: “have” -> has

We have made the suggested correction (Line 166).

Line 173: “model” -> models

We think we should use the singular subject 'model' there.

Line 184: “its spatial patterns clearly exhibit” -> its spatial pattern clearly exhibit

We have made the suggested correction (Line 179).

Line 190: “may physically connected” -> may be physically connected

We have made the suggested correction (Line 185).

Line 194: “in climatology” -> in the climatology

We have made the suggested correction (Line 189).

Fig. 2c: The x-axis label is missing. Should be “Days” as in the Supplement.

Thank you very much! Added (Figure 2, Line 801).

Line 251: “The similar” -> A similar

We have made the suggested correction (Line 245).

Line 262: Please add that Arctic is north of 60°N here.

We have made the suggested correction (Line 258).

Line 272: “that large-scale circulation” -> that the large-scale circulation

We have made the suggested correction (Line 268).

Line 278: “to top of” -> to the top of

We have made the suggested correction (Line 275).

Line 297-298: Remove “, which is characterized by a zonal wave number 2 structure,”

We have made the suggested correction (Line 294).

Line 300: Where is “the high pressure”? Over Canada/Canadian archipelago?

We referred to the high-pressure anomalies over Greenland and central Siberia. We have added a few words there to clarify this point (Line 297).

Lines 302-303: Is the response of northern Europe really above 0.5 days per month?

The AR response over northern Europe is around 0.5 days per month. To make the description more accurate, we have modified the sentences (Line 299).

Lines 336, 346, 347, 365: “AR”: AR frequency

We have made the suggested correction (Lines 333, 343, 345, and 362).

Line 400: “6-hour” -> 6-hourly

We have made the suggested correction (Lines 395, 398).

Line 470: “spatiotemporally” -> spatially

We have made the suggested correction (Line 472).

Lines 474, 475: “grids” -> grid points

We have made the suggested correction (Lines 476, 477).

Caption of Figure 1:

Lines 783, 788, 789: “ARs” -> ARs frequency

We have made the suggested correction (Lines 787, 792, and 794).

“lower and middle” -> lower to middle

We have made the suggested correction (All figure captions containing 'lower and middle').

Lines 785-786: remove “, or m/s/10a”

We have made the suggested correction (Line 795).

Line 789: “denotes” -> denote

We have made the suggested correction (Line 797).

Line 790: “for wind” -> for the wind

We have made the suggested correction (Lines 790-804).

Caption of Figures 4 and 5:

Line 829, 839, 840: “ARs” -> AR frequency

We have made the suggested correction (Lines 827, 834, 840, 841, and 843).

Please also check the captions of the Supplement Figures.

Thanks for your suggestion! We have double-checked the figure captions for the supplementary figures and have corrected all existing typos and errors.

REVIEWER COMMENTS

Reviewer #2 (Remarks to the Author):

The authors have rather satisfactorily answered my comments and I believe the manuscript was improved. I still have one major comment along with minor issues listed here below. I recommend that the authors make the suggested modifications before acceptance of the manuscript for publication.

Major comment:

Figure 1g, 2d, and Supplementary Figures 7c, 8c: Following my comments on what is represented in these panels, the authors have changed the y-axis label from “Standard deviation” to “Standardized time coefficient”. This latter is better but still not precise to my opinion. If the authors were using EOFs, they would represent in these plots the time series of the principal component of EOF1. Please correct me if I have misunderstood what is actually represented here. However, the authors use the Maximum Covariance Analysis and the principal component is now called expansion coefficient or time expansion coefficient as for example mentioned in An (2003). Therefore, I strongly suggest to the authors to use “Expansion coefficient” or “Standardized expansion coefficient”.

Technical and minor comments:

Line 100: It seems that the paper linked to reference 56 deals mostly with actual rivers over land and not much with atmospheric rivers. Are you sure that this citation is appropriate here?

Line 125: “regulating moisture” → regulating the moisture

Supplementary Figure 3: the labels of the color bars should just contain the variable name, that is remove the word “mean”. Also “AR mean” should AR frequency, as in the caption.

Line 142: “northeastern” → northern. It looks to me that the whole Arctic side of Canada experience an increase in AR frequency.

Lines 199-200: “we use the time series of MCA1-ARs (or MCA1-Z200) to correlate with detrended JJA low cloud” → we correlate the time series of MCA1-ARs (or MCA1-Z200) with the detrended JJA low cloud

Line 218-219: “our first examination centers in an analysis of that year” → our first examination centers on that year

Line 219-220: “MCA analysis [...] are computed” → An MCA analysis [...] is performed

Figure 1: Panel (a) shows that the atmospheric river pathway into the Arctic west of Greenland will be strengthened with more ARs in this region. However, the trend in cyclone frequency is quite weak as seen in Fig. 3f in Zhang et al. (2023) (<https://www.nature.com/articles/s43247-023-01003-0/figures/3>) and not significant. Moreover, the cyclone frequency seems also to increase over the Nordic Seas (Greenland, Norwegian, and Barents Seas) although the AR frequency trend the authors find is negative. How do the authors explain these two different behaviors? This could maybe be included in the discussion section.

References:

- An, S.-I. (2003) Conditional Maximum Covariance Analysis and Its Application to the Tropical Indian Ocean SST and Surface Wind Stress Anomalies. *J. Climate*, 16, 2932–2938. [https://doi.org/10.1175/1520-0442\(2003\)016<2932:CMCAAI>2.0.CO;2](https://doi.org/10.1175/1520-0442(2003)016<2932:CMCAAI>2.0.CO;2)
- Zhang, X. et al. Arctic cyclones have become more intense and longer-lived over the past seven decades. *Commun Earth Environ* 4, 348 (2023). <https://doi.org/10.1038/s43247-023-01003-0>
- Zhang, Z., Ralph, F. M., & Zheng, M. (2019). The relationship between extratropical cyclone strength and atmospheric river intensity and position. *Geophysical Research Letters*, 46, 1814–1823. <https://doi.org/10.1029/2018GL079071>

REVIEWER COMMENTS

Reviewer #2 (Remarks to the Author):

The authors have rather satisfactorily answered my comments and I believe the manuscript was improved. I still have one major comment along with minor issues listed here below. I recommend that the authors make the suggested modifications before acceptance of the manuscript for publication.

We are grateful to the reviewer again for the valuable suggestions, which offer us another opportunity to improve our manuscript. We hope the changes we have made in response to the reviewer's feedback meet the expectations.

Major comment:

Figure 1g, 2d, and Supplementary Figures 7c, 8c: Following my comments on what is represented in these panels, the authors have changed the y-axis label from “Standard deviation” to “Standardized time coefficient”. This latter is better but still not precise to my opinion. If the authors were using EOFs, they would represent in these plots the time series of the principal component of EOF1. Please correct me if I have misunderstood what is actually represented here. However, the authors use the Maximum Covariance Analysis and the principal component is now called expansion coefficient or time expansion coefficient as for example mentioned in An (2003). Therefore, I strongly suggest to the authors to use “Expansion coefficient” or “Standardized expansion coefficient”.

Thank you for your suggestion. We have replaced “Standardized time coefficient” with “Expansion coefficient” in Figure 1g, 2d, and Supplementary Figures 7c, 8c, 9c.

Technical and minor comments:

Line 100: It seems that the paper linked to reference 56 deals mostly with actual rivers over land and not much with atmospheric rivers. Are you sure that this citation is appropriate here?

That paper primarily discusses the impact of human activities on river runoff, but it also analyzes the contribution of ARs to changes in river runoff. It was found that the contribution of ARs to changes in the flow intensity of some rivers, such as the Amur, Zhujiang (Pearl), and Columbia, exceeded 30%. Therefore, we believe it is appropriate to cite that work to support our main point of that sentence.

Line 125: “regulating moisture” → regulating the moisture

We have made the suggested correction (Line 125).

Supplementary Figure 3: the labels of the color bars should just contain the variable name, that is remove the word “mean”. Also “AR mean” should AR frequency, as in the caption.

We thank the reviewer for this suggestion. We have replaced “AR mean” with “AR frequency” in Supplementary Figure 3a.

Line 142: “northeastern” → northern. It looks to me that the whole Arctic side of Canada experience an increase in AR frequency.

We have made the suggested correction (Line 142).

Lines 199-200: “we use the time series of MCA1-ARs (or MCA1-Z200) to correlate with detrended JJA low cloud” → we correlate the time series of MCA1-ARs (or MCA1-Z200) with the detrended JJA low cloud

We have made the suggested correction (Lines 199-200).

Line 218-219: “our first examination centers in an analysis of that year” → our first examination centers on that year

We have made the suggested correction (Line 219).

Line 219-220: “MCA analysis [...] are computed” → An MCA analysis [...] is performed

Thanks for your suggestion. We have made the suggested correction (Lines 219-220).

Figure 1: Panel (a) shows that the atmospheric river pathway into the Arctic west of Greenland will be strengthened with more ARs in this region. However, the trend in cyclone frequency is quite weak as seen in Fig. 3f in Zhang et al. (2023) (<https://www.nature.com/articles/s43247-023-01003-0/figures/3>) and not significant. Moreover, the cyclone frequency seems also to increase over the Nordic Seas (Greenland, Norwegian, and Barents Seas) although the AR frequency trend the authors find is negative. How do the authors explain these two different behaviors? This could maybe be included in the discussion section.

Thanks for pointing us to this paper. In Zhang et al. (2023), the authors compared the frequency of strong cyclones between two epochs (1950-1985 vs 1986-2021), which

is different from our timeframe focusing on long-term changes in AR frequency from 1979 to 2019. Thus, the two papers examine changes in two different variables over different periods. Additionally, it is known that the occurrence of ARs requires two key elements: strong cyclonic activity and an abundance of moisture from lower latitudes. In Zhang et al. (2023), the main criterion to determine strong cyclones is the presence of an isolated low sea level pressure and the strength of geostrophic winds associated with cyclones. The influence of moisture is not considered in their detection method. Given the different physical emphasis of our AR detection approach and their cyclone detection method, and the different time frames between the two studies, we can't expect that the long-term change in two fields should resemble each other. Lastly, I would like to use the key conclusion of another study by Zhang et al. (2019) to reinforce this point : in extratropical regions, ARs and cyclones do not always coexist. Therefore, the occurrence of ARs cannot be simply determined by the presence of cyclones.

References:

Zhang, Z., Ralph, F. M., & Zheng, M. (2019). The relationship between extratropical cyclone strength and atmospheric river intensity and position. *Geophysical Research Letters*, 46, 1814 – 1823. <https://doi.org/10.1029/2018GL079071>